# Topological obstruction to the training of shallow ReLU neural networks

**Marco Nurisso**
Politecnico di Torino & CENTAI Institute
Torino, 10100 - ITALY
`marco.nurisso@polito.it`

**Pierrick Leroy**
Politecnico di Torino
Torino, 10100 - ITALY
`pierrick.leroy@polito.it`

**Francesco Vaccarino**
Politecnico di Torino
Torino, 10100 - ITALY
`francesco.vaccarino@polito.it`

## Abstract

Studying the interplay between the geometry of the loss landscape and the optimization trajectories of simple neural networks is a fundamental step for understanding their behavior in more complex settings. This paper reveals the presence of topological obstruction in the loss landscape of shallow ReLU neural networks trained using gradient flow. We discuss how the homogeneous nature of the ReLU activation function constrains the training trajectories to lie on a product of quadric hypersurfaces whose shape depends on the particular initialization of the network's parameters. When the neural network's output is a single scalar, we prove that these quadrics can have multiple connected components, limiting the set of reachable parameters during training. We analytically compute the number of these components and discuss the possibility of mapping one to the other through neuron rescaling and permutation. In this simple setting, we find that the non-connectedness results in a topological obstruction, which, depending on the initialization, can make the global optimum unreachable. We validate this result with numerical experiments.

## 1   Introduction

Training a neural network consists of navigating the complex geometry of the loss landscape to reach one of its deepest valleys. Gradient descent and its variants are, by far, the most commonly used algorithms to perform this task. While technically correct, the standard picture of the parameter space as Euclidean space with the trajectory rolling down the loss's surface in the steepest direction towards a minimum is slightly misleading because different choices of parameters can be *observationally equivalent* i.e. encode the same function [10]. The observational equivalence of parameters shape the loss landscape by imposing specific geometric structures on the parameter space. Minima are not isolated points but high-dimensional manifolds with complex geometry [17, 9, 42] and the loss function's gradients and Hessian are constrained to obey some specific laws [46, 27]. Gradient-based optimization methods, where the parameters are updated by performing discrete steps in the gradient's direction, are thus very much dependent on the symmetry-induced geometry [12, 29].

In this work, we provide a topological perspective on the constraints induced by some groups of network symmetries on the optimization trajectories. Topology is a field of mathematics that studies the properties of a space that are preserved under continuous deformations. Our main goal is to find and quantify in topological terms the impossibility of the training trajectories to freely explore the parameter space and get from any initialization to an optimal parameter. This idea is formalized in

38th Conference on Neural Information Processing Systems (NeurIPS 2024).

the topological notion of *connectedness* and, in particular, with the 0-th *Betti number*, which counts the number of *connected components* the space is composed of. The presence, or the absence, of topological *obstructions* in the parameter space does not depend on the particular loss function or the training data but is intrinsic to the interplay between the geometry and the topology of the parameter space under the action of groups of symmetries inducing observationally equivalent networks.

**Main contributions.** Our main contributions are the following.

1. We find that, for two-layer neural networks, the gradient flow trajectories lie on an invariant set, which can be factored as the product of quadric hypersurfaces.
2. We analytically compute its Betti numbers, i.e., the number of connected components, holes, and higher-dimensional cavities.
3. We find that the invariant set can be disconnected when the network's output dimension is 1, leading to a clear topological obstruction.
4. We find that the obstruction is caused by "pathological" neurons that cannot change the sign of their output weights when trained with gradient flow.
5. We discuss the relation between the invariant set and the network's symmetries, finding that if we consider permutations, the number of effective connected components scales linearly in the number of pathological neurons.
6. We perform numerical validations on controlled toy scenarios, displaying the effect of obstruction in practice.

## 2 Related work

A large body of work studies gradient flow and gradient descent optimization of one hidden layer networks with homogeneous activations. Convergence properties have been found for wide networks [37, 43] with bounded density at initialization [31]. The implicit regularization provided is studied under various assumptions on: orthogonal input data [4], initialization scale [30, 4], wide (overparameterized regime) and infinitely wide [6], linearly separable data [30, 45]. Deeper linear networks [24] have also been studied.

These works focus on proving convergence and understanding which (optimal) solution is found, whereas our work investigates the shape of the optimization space and focuses on cases where the optimum might not be reachable from a given initialization.

Closer to our work, Safran et al. [39] studies two-layer ReLU binary classifiers with single input and output, counting the number of their piecewise-linear components after training. Eberle et al. [13] focuses on the differential challenge posed by the ReLU activation function and studies properties like the uniqueness of the solution of a gradient flow differential equation for a given initialization.

ReLU activation is a nonnegative homogeneous function, meaning that particular weight rescalings do not change the neural network's function. This is at the heart of the counterargument to flatness measures made by Dinh et al. [10], which shows that the Hessian eigenvalues can be made arbitrarily large in this way. Neyshabur et al. [34] explores the effect such rescalings can have on the gradient, proposing a rescaling-invariant regularization, and Pittorino et al. [36] employs them to define invariant flatness measures. Generally speaking, neural networks possess symmetries [20], and symmetries influence the geometry of training. Du et al. [12] studies how symmetry leads ReLU networks to automatically balance the neurons' weights. Kunin et al. [27], Zhao et al. [51] study how it constrains the gradient and Hessian matrix, leading to conservation laws w.r.t. gradient flow and Tanaka et al. [46] leverages it to propose a network pruning scheme. Ziyin [52] studies general mirror-reflect symmetries of the loss function and their effect on the weights of the trained network. Other conserved quantities stem from batch normalization's scale invariance [23, 47]. The transition from gradient flow to finite step size gradient descent breaks the conservation laws, resulting in altered trajectories [14, 2, 27, 44].

Numerous works have explored the geometry and topology of the loss landscape to obtain insight into a neural network's training behavior. Motivated by the striking experimental observation that low loss points can be connected by simple curves [11, 18] or line segments [40, 16, 15], a large body of literature tries to understand this phenomenon of mode connectivity under the topological lens of the connectedness of the loss function's sublevel sets [17, 35, 26], especially for overparameterized neural networks [9, 8, 42]. Another line of work approaches the connectivity of minima from another point of view, studying the presence [50, 38, 48] or absence [28] of spurious minima, i.e. minima

which are not global. Bucarelli et al. [5] analytically derives bounds on the sum of the Betti numbers of the loss landscape's sublevel set. Topological data analysis methods have also been exploited to numerically study the shape of the loss landscape [1, 22].

## 3 Setup and preliminaries

### 3.1 One-hidden layer neural network

Unless otherwise stated, all vectors are column vectors, that is, $x = (x_1, \ldots, x_d)^\top \in \mathbb{R}^d \cong \mathbb{R}^{d \times 1}$. Let us consider a two-layer neural network $f(\cdot, \theta) : \mathbb{R}^d \to \mathbb{R}^e$ specified by the function

$$f(x; \theta) = W^{(2)} \sigma(W^{(1)} x), \tag{1}$$

where $x \in \mathbb{R}^d$ is the input, $\theta = (W^{(1)}, W^{(2)})$ with $W^{(1)} \in \mathbb{R}^{l \times d}$ and $W^{(2)} \in \mathbb{R}^{e \times l}$ are the parameters, $\sigma : \mathbb{R} \to \mathbb{R}$ is the component-wise activation function and $l$ is the number of neurons in the hidden layer. Notice that we consider a network with no biases, as it allows us a discussion with lighter notation. The case with biases is discussed in Appendix E.

In this work, following [12], we focus on the case where $\sigma$ is *homogeneous*, namely $\sigma(x) = \sigma'(x) \cdot x$ for every $x$ and for every element of the sub-differential $\sigma'(x)$ if $\sigma$ is non-differentiable at $x$. The commonly used ReLU ($\sigma(z) = \max\{z, 0\}$) and Leaky ReLU ($\sigma(z) = \max\{z, \gamma\}$ with $0 \le \gamma \le 1$) activation functions satisfy this property.

We call *parameter space* the vector space $\Theta = \left\{ \theta = (W^{(1)}, W^{(2)}) \mid W^{(1)} \in \mathbb{R}^{l \times d}, W^{(2)} \in \mathbb{R}^{e \times l} \right\}$.

It will also be convenient to examine the single hidden neurons and their associated parameters for the following discussions.

**Proposition 1.** *For the two-layer neural network defined in Equation* (1). *Let* $k = 1, \ldots, l$, *let* $(e_{11}, e_{12}, \ldots, e_{ll})$ *be the canonical basis of* $\mathbb{R}^{l \times l}$ *and* $\Theta_k = \left\{ \theta_k = (e_{kk} W^{(1)}, W^{(2)} e_{kk}) \mid (W^{(1)}, W^{(2)}) \in \Theta \right\} \subset \Theta$, *then* $\Theta = \Theta_1 \oplus \cdots \oplus \Theta_l$.

Details of the proof are provided in Appendix A. Fixing $k \in \{1, \ldots, l\}$, we can consider $\Theta_k$ as the parameter space of the $k$-th hidden neuron, which consists of the inputs and output weights of neuron $k$, namely the rows and columns of $W^{(1)}$ and $W^{(2)}$, respectively. For simplicity, when we work in $\Theta_k$, we write $W_k^{(1)} := e_{kk} W^{(1)}$ and $W_k^{(2)} := W^{(2)} e_{kk}$. Interestingly, the decomposition of Proposition 1 only holds for two-layer neural networks and will be crucial to the formulations of this paper's results.

### 3.2 Symmetries and observationally equivalent networks

It is well known that the properties of the activation function heavily influence the geometry of the parameter space $\Theta$. The activation function's commutativity with some classes of transformations can result in the latter having no effect on the function implemented by the neural network. This means that, in general, the mapping from the parameter space to the hypothesis class of functions is not injective. Following the terminology in Dinh et al. [10], we say that two parameters $\theta_1, \theta_2 \in \Theta$ are *observationally equivalent*, if they encode the same function $f(\cdot; \theta_1) = f(\cdot, \theta_2)$ and write $\theta_1 \sim \theta_2$.

In the case of homogeneous activations (ReLU or Leaky ReLU), we describe two kinds of transformations that send a parameter $\theta$ into an observationally equivalent one.

**Neuron rescaling.** The input weights of a hidden neuron can be rescaled by a positive scalar $\alpha > 0$ provided that its output weights are rescaled by the inverse $\alpha^{-1}$ (top panel of Figure 1a). We formalize this as the action of the group $\mathbb{R}_+$ of positive real numbers on $\Theta_k$:

$$T : \mathbb{R}_+ \times \Theta_k \to \Theta_k \tag{2}$$
$$(\alpha, \theta_k) \mapsto T_\alpha(\theta_k) = \left( \alpha \cdot W_k^{(1)}, \frac{1}{\alpha} \cdot W_k^{(2)} \right).$$

This action can be naturally extended to the space of all parameters by considering the possibility of rescaling all hidden neurons simultaneously by different factors. If $\alpha = (\alpha_1, \ldots, \alpha_l) \in \mathbb{R}_+^l$

$$T_\alpha(\theta) = (\mathrm{diag}(\alpha) W^{(1)}, W^{(2)} \mathrm{diag}(\alpha)^{-1}). \tag{3}$$

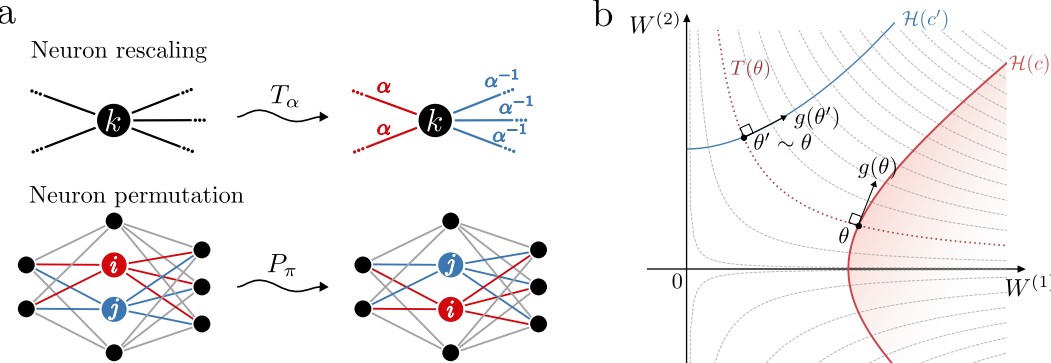

Figure 1: **a.** Depiction of the two group actions acting on the space of the network's parameters: the neuron rescaling of Equation (2) (top) and the neuron permutation of Equation (4) (bottom). **b.** Depiction of the geometry of the parameter space induced by the rescaling invariance of ReLU networks. The dotted lines denote the orbits $T(\theta)$ while the solid lines represent the invariant sets $\mathcal{H}(c)$ associated with $\theta$ and the one associated with its rescaled version $\theta'$. Notice how the gradient of the loss $g(\theta)$ is tangent to $\mathcal{H}(c)$ and orthogonal to $T(\theta)$.

Given that $\sigma(az) = a\sigma(z)$ when $a \in \mathbb{R}_+$, we see how $\theta \sim T_\alpha(\theta)$.

We write $T(\theta)$ to denote the orbit of a parameter $\theta$ under the action of $T$, i.e. the set of all parameters obtained from $\theta$ by arbitrarily rescaling the neurons $T(\theta) = \{T_\alpha(\theta) : \alpha \in \mathbb{R}_+^l\}$.

**Permutations of the neurons.** Besides rescaling, we can obtain an observationally equivalent network by permuting the hidden neurons in such a way as to preserve their input and output weights (bottom panel of Figure 1a).

Given the symmetric group on $l$ elements $\mathfrak{S}_l$ of the permutations of $\{1, \ldots, l\}$, we write the action

$$P\colon \mathfrak{S}_l \times \Theta \to \Theta \tag{4}$$
$$(\pi, \theta) \mapsto P_\pi(\theta) = (R_\pi W^{(1)}, W^{(2)} R_\pi^\top).$$

where $R_\pi$ is the $l \times l$ row-permutation matrix associated to the permutation $\pi$.

Given that the activation function $\sigma$ is applied component-wise, we have that it commutes with $R_\pi$, namely

$$f(x; P_\pi(\theta)) = W^{(2)} R_\pi^\top \sigma(R_\pi W^{(1)} x) = W^{(2)} R_\pi^\top R_\pi \sigma(W^{(1)} x) = W^{(2)} \sigma(W^{(1)} x) = f(x; \theta)$$

and thus $P_\pi(\theta) \sim \theta$ because $R_\pi^\top = (R_\pi)^{-1}$.

Having defined these two actions, we say that $\theta$ and $\theta'$ are *observationally equivalent by rescalings and permutations* if $\theta'$ can be obtained from $\theta$ by a finite sequence of actions of $T$ and $P$ or, equivalently thanks to Lemma 3 in the Appendix, if there exists a rescaling $\alpha$ and a permutation $\pi$ such that $\theta' = P_\pi \circ T_\alpha(\theta)$. In this case, we write $\theta \stackrel{\text{rp}}{\sim} \theta'$.

### 3.3 Conserved quantities and the invariant hyperquadrics

The presence of symmetries in the neural network's parameter-function map results in a specific geometric structure in the loss landscape. Let indeed $D = \{(x_i, y_i) \in \mathbb{R}^d \times \mathbb{R}^e\}_{i=1}^N$ be a training set of $N$ input-output pairs and fix a loss function $L : \Theta \to \mathbb{R}$ which depends on the parameters only through the output of the neural network (1), that is

$$L(\theta) = \frac{1}{N} \sum_{i=1}^N \ell(f(x_i; \theta), y_i) \tag{5}$$

where $\ell : \mathbb{R}^e \times \mathbb{R}^e \to \mathbb{R}$ is differentiable. In this work, as empirical risk minimization, we consider the continuous time version of the gradient descent (GD) algorithm (with learning rate $h > 0$)

$$\theta_{t+1} = \theta_t - h \nabla_\theta L(\theta_t) \tag{6}$$

named *gradient flow* (GF), and defined as

$$\frac{\mathrm{d}}{\mathrm{d}t}\theta(t) \in -\nabla_\theta L(\theta(t)) := -g(\theta(t)) \tag{7}$$

where $\nabla_\theta L(\theta(t))$ is the Clarke sub-differential [7] which takes into account the parameters $\theta$ where $L(\theta)$ is non-differentiable. Given that the loss function $L$ depends on the parameters only through $f$, its value at $\theta$ must be constant over the orbit $T(\theta)$. This, together with the fact that the gradient of a differentiable function at a point is orthogonal to the level set at that point, means that

$$g(\theta) \perp T(\theta) \tag{8}$$

at any parameter $\theta$ where $L(\theta)$ is differentiable, as represented in Figure 1b. This orthogonality condition constrains the possible values of the gradient and, by extension, the possible gradient flow trajectories. In particular, as proven in Liang et al. [29], Tanaka et al. [46], Equation (8) is equivalent to

$$\sum_{i=1}^{d} W_{ki}^{(1)} g_{ki}^{(1)} - \sum_{j=1}^{e} W_{jk}^{(2)} g_{jk}^{(2)} = 0 \quad \forall k = 1, \dots, l. \tag{9}$$

For convenience of notation, we define, for $k = 1, \dots, l$, the following bilinear forms on $\Theta$, which help us describe the geometry induced by the rescaling symmetry. If $\theta = (W^{(1)}, W^{(2)})$ and $\eta = (V^{(1)}, V^{(2)})$, we define

$$\langle\!\langle \theta, \eta \rangle\!\rangle_k = \sum_{i=1}^{d} W_{ki}^{(1)} V_{ki}^{(1)} - \sum_{j=1}^{e} W_{jk}^{(2)} V_{jk}^{(2)} \tag{10}$$

which, notice, only depends on the $k$-th row of $W^{(1)}$ and $k$-th column of $W^{(2)}$ meaning that we can equivalently see it as a bilinear form on $\Theta_k$. $\Theta_k$, together with $\langle\!\langle \cdot, \cdot \rangle\!\rangle_k$ is a *pseudo-Euclidean space*.

With the notation given by Equation (10), we see that Equation (9) can be simply rewritten as $\langle\!\langle \theta, g(\theta) \rangle\!\rangle_k = 0$ for every neuron $k$. This condition, akin to orthogonality w.r.t. the bilinear form of Equation (10), implies that, under gradient flow optimization,

$$\frac{\mathrm{d}}{\mathrm{d}t} \langle\!\langle \theta, \theta \rangle\!\rangle_k = 2\langle\!\langle \dot\theta, \theta \rangle\!\rangle_k = -2\langle\!\langle g(\theta), \theta \rangle\!\rangle_k = 0 \quad \forall k = 1, \dots, l. \tag{11}$$

This result, first obtained in Saxe et al. [41] for linear networks and discussed in Du et al. [12], Liang et al. [29], Kunin et al. [27], tells us that the rescaling symmetry results in the quantities $\langle\!\langle \theta, \theta \rangle\!\rangle_k$ being conserved. This means that the difference between the Euclidean norm of the inputs and the outputs is constant for each neuron throughout the GF training trajectory. Moreover, under the condition of homogeneity of the activation function, Du et al. [12] proves that Equation (11) holds even at non-differentiable points of $L$ and in the case of multiple layers.

**Invariant sets.**   Assume that at the initialization $\theta_0$ we have $\langle\!\langle \theta_0, \theta_0 \rangle\!\rangle_k = c_k$, for all $k$, then Equation (11) implies that the GF trajectory will lie on the set characterized by the system of equations $\langle\!\langle \theta, \theta \rangle\!\rangle_k = c_k$ for $k = 1, \dots, l$. This subset is mapped to itself under the GF dynamics by Equation (11) (see Figure 1b) and constitutes the main object of our study.

**Definition 1** (Invariant set). *Given $c = (c_1, \dots, c_l)$, we call* invariant set *the subset $\mathcal{H}(c) \subseteq \Theta$ given by the equations $\langle\!\langle \theta, \theta \rangle\!\rangle_k = c_k \ \forall k = 1, \dots, l$.*

If we look at each single equation (i.e. to each hidden neuron), we see that Equation (11) can be written as

$$\sum_{i=1}^{d} \left( W_{ki}^{(1)} \right)^2 - \sum_{j=1}^{e} \left( W_{jk}^{(2)} \right)^2 = c_k \tag{12}$$

which corresponds to a *hyperquadric* (or quadric hypersurface) in $\Theta_k$. We denote with $\mathcal{Q}(c_k) \subseteq \Theta_k$ this hypersurface and call it the *invariant hyperquadric* associated to the $k$-th hidden neuron.

Here $c_k \in \mathbb{R}$ takes the role of a label associated with the $k$-th hidden neuron, which, we see in the next section, plays a key role in specifying the shape of $\mathcal{Q}(c_k)$. Figure 2a shows how, for $d = 2$ and $e = 1$, $\mathcal{Q}(c_k)$ is an hyperboloid with 1 sheet (connected) if $c_k > 0$ and 2 sheets if $c_k < 0$.

# 4 Topology of the invariant set

As we discussed above, Equation (11) tells us that gradient flow trajectories can't explore the whole space $\Theta$ but are constrained to lie on the invariant set $\mathcal{H}(c)$. The values of $c$, in turn, depend on the initialization and, we see from Equation (12), quantify the balance between the norms of input and output weights in every hidden neuron.

The goal of this section is to provide a topological characterization of $\mathcal{H}(c)$ that can tell us something about the presence or absence of fundamental *obstructions* to the network's training process. With obstruction, we mean the impossibility of a GF trajectory to travel freely from one point to the other in $\mathcal{H}(c)$. We refer the reader to Appendix B for an essential overview of some of the topological concepts that we rely on in the next paragraphs.

**Counting high-dimensional holes.** Our topological characterization will be framed using *Betti numbers*. Betti numbers are well-known topological invariants given by a sequence of natural numbers that intuitively encode the number of higher-dimensional holes and cavities present in space. In particular, the 0-th Betti number of a space $X$, $\beta_0(X)$ corresponds to the number of connected components of $X$ and thus will be fundamental for our goal of identifying obstructions.

The invariant set $\mathcal{H}(c)$ is given as the set of solutions of $l$ polynomial equations of degree 2 sharing no variables. Furthermore, in the setting of two-layer neural networks, we can leverage the fact that the parameter space can be decomposed into the parameter spaces of the hidden neurons. This, in turn, allows us to decompose the invariant set as the product of the neurons' invariant hyperquadrics, greatly simplifying our study.

**Lemma 1.** *In a two-layer ReLU neural network, the invariant set $\mathcal{H}(c)$ is homeomorphic to the Cartesian product of the hidden neurons' invariant hyperquadrics, that is*

$$\mathcal{H}(c) \cong \mathcal{Q}(c_1) \times \cdots \times \mathcal{Q}(c_l). \tag{13}$$

Lemma 1 tells us that we can understand the topology of $\mathcal{H}(c)$ by studying independently its factors. Moreover, the hyperquadrics we encounter here are well-studied objects for which the next proposition (proven in Appendix D.2) gives a topological characterization.

**Proposition 2.** *If $c_k > 0$, $\mathcal{Q}(c_k)$ is a topological manifold homeomorphic to $\mathbb{R}^e \times S^{d-1}$. If $c_k < 0$, $\mathcal{Q}(c_k)$ is a topological manifold homeomorphic to $\mathbb{R}^d \times S^{e-1}$. If $c_k = 0$, $\mathcal{Q}(0)$ is a contractible space.*

Leveraging the decomposition of Lemma 1 and the characterization of the factors given by Proposition 2, we can explicitly compute all the Betti numbers of the invariant set. We give the next result in terms of the *Poincaré polynomial* of $\mathcal{H}(c)$, namely the polynomial whose coefficients are the Betti numbers (see Appendix B).

**Theorem 1.** *Let $l_+, l_-, l_0$ be the number of positive, negative, and zero components of $c$, respectively. The Poincaré polynomial of $\mathcal{H}(c)$ is given by*

$$p_{\mathcal{H}(c)}(x) = (1 + x^{d-1})^{l_+}(1 + x^{e-1})^{l_-} \tag{14}$$

This result, which is proven in Appendix D.3, contains a wealth of topological information as it gives us the exact number of holes and cavities of any order, depending on the network's hyperparameters $(d, e)$ and initialization $(l_+, l_-)$. In the rest of this work, we focus only on the 0-th Betti number as the non-connectedness of $\mathcal{H}(c)$ provides a clear obstruction to the GF trajectories.

**Connectedness of the invariant set.** With regard to the connectedness of $\mathcal{H}(c)$, we can leverage Theorem 1 to obtain the exact number of connected components.

**Corollary 1.** *The 0-th Betti number $\beta_0$ of $\mathcal{H}(c)$, corresponding to the number of its connected components, is given by*

$$\beta_0 = \begin{cases} 1 & \text{if } d, e > 1 \\ 2^{l_+} & \text{if } d = 1, e > 1 \\ 2^{l_-} & \text{if } d > 1, e = 1 \\ 2^{l_+ + l_-} & \text{if } d = 1, e = 1 \end{cases} \tag{15}$$

*Proof.* This can be directly obtained from the coefficient of degree 0 of the Poincaré polynomial obtained through Theorem 1. $\square$

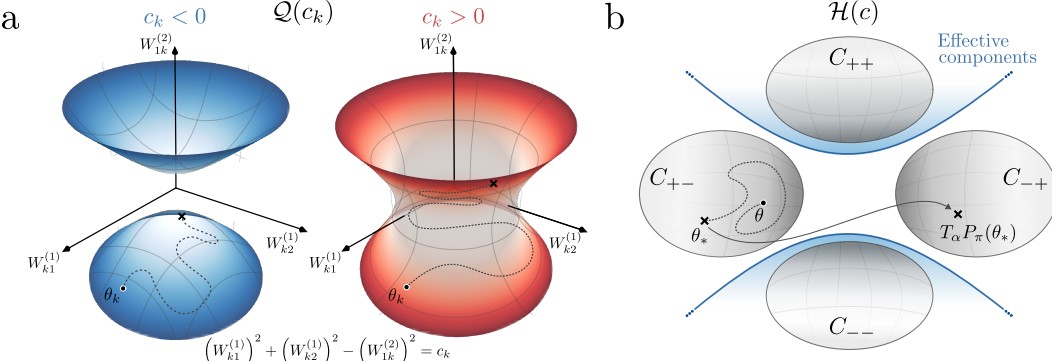

Figure 2: **a.** The invariant hyperquadric $\mathcal{Q}(c_k)$ of a neuron with two inputs ($d = 2$) and one output ($e = 1$) in the cases where $c_k < 0$ (left) and $c_k > 0$ (right). **b.** Depiction of the invariant set $\mathcal{H}(c)$ in the case where $l_- = 2$ so that there are $2^{l_-} = 4$ connected components. $C_{\pm\mp}$ denotes the connected component such that $s = (\pm 1, \mp 1)$. The blue lines separate the different effective components of $\mathcal{H}(c)$.

What we see in Equation (15) is that in most cases, the invariant set is connected, and gradient flow has no topological limitations in exploring the whole of $\mathcal{H}(c)$. Instead, when the hidden neurons have only one input or only one output, the space is fragmented into several components whose number scales exponentially in $l_+$ or $l_-$, respectively.

Let us focus on the more interesting case where $d > 1$ and $e = 1$.

**Corollary 2.** *If the output of a two-layer ReLU neural network is a single scalar $e = 1$, its input has dimension $d > 1$, and the initial parameter $\theta_0$ is such that $\langle\!\langle \theta_0, \theta_0 \rangle\!\rangle_k < 0$ for $l_- > 0$ hidden neurons, then the set $\mathcal{H}(c)$ is disconnected and has $2^{l_-}$ connected components.*

This means that neurons initialized with the norm of their outgoing weight strictly greater than their incoming weights' norm are responsible for disconnecting the space. We now precisely identify which connected component a parameter $\theta$ belongs to and clarify the meaning of the obstruction.

**Proposition 3.** *Let $e = 1, d > 1$, and $\theta \in \mathcal{H}(c)$ with $c$ such that $c_{k_1}, \ldots, c_{k_{l_-}} < 0$ while $c_k \geq 0$ for all other $k$. Let $W_-^{(2)} := (W_{k_1}^{(2)}, \ldots, W_{k_{l_-}}^{(2)}) \in \mathbb{R}^{1 \times l_-}$ be the row vector whose components are the components of $W^{(2)} \in \mathbb{R}^{1 \times l}$ associated to $c_k < 0$. Then the vector $s(\theta) = (\mathrm{sign}(W_{k_1}^{(2)}), \ldots, \mathrm{sign}(W_{k_{l_-}}^{(2)}))$ identifies uniquely the component $\theta$ belongs to, namely: $\theta$ and $\theta'$ belong to the same connected component of $\mathcal{H}(c)$ if and only if $s(\theta) = s(\theta')$.*

Proposition 3, proven in Appendix D.4, implies that $s(\theta)$ does not change when we move in $C$ on a continuous curve such as the one given by gradient flow. This gives us an interesting interpretation of the topological obstruction: gradient flow cannot change the signs of the outgoing weights of the hidden neurons $k$ such that $c_k < 0$ (see Appendix G for an intuitive explanation of the phenomenon). This same observation is also mentioned in Boursier and Flammarion [3]. Proposition 3 extends one of the results of Boursier et al. [4] which proves that the same also holds when $c_k = 0$ (balanced initialization).

By also considering Corollary 1, one obtains that a clever initialization of the parameters given by $\langle\!\langle \theta_0, \theta_0 \rangle\!\rangle_k = c_k > 0 \ \forall k = 1, \ldots, l$ can prevent the issue by ensuring the connectedness of the invariant set. We also find that under common initialization schemes such as Xavier [19] and Kaiming [21] the probability of having pathological neurons is negligible when the input dimension and number of hidden neurons is high (see Appendix F).

## 5  Taking symmetries into account

Corollary 2 states that neurons $k$ such that $c_k < 0$ are "pathological", in the sense that they are responsible for disconnecting the invariant set into several components, whose number scales exponentially in the number of those neurons. This result gives us a grim picture of the possibility of actually

optimizing the neural network: if the initial parameter $\theta_0$ is in a particular connected component and the global optimum $\theta_*$ lies in another, then any gradient flow trajectory will not be able to reach $\theta_*$ because it will be constrained in its connected component.

This result, however, provides us only with a partial picture of the parameter space's geometry. It is a priori possible that the training trajectory, moving in its connected component, reaches a parameter $\zeta$, which itself is optimal as it is observationally equivalent to $\theta_*$ ($\zeta \sim \theta_*$). In this case, the topological obstruction given by the non-connectedness would be only apparent.

To take this fact into account, we define the following notion.

**Definition 2** (Effective component). *Let $\theta \in \mathcal{H}(c)$ and $C(\theta)$ be its connected component therein. We define its effective component $\mathrm{Eff}(\theta)$ as the union of the connected component of all $\theta'$ such that $\theta' \overset{\mathrm{rp}}{\sim} \theta$. So that $\mathrm{Eff}(\theta) := \bigcup_{\theta' \overset{\mathrm{rp}}{\sim} \theta} C(\theta')$.*

Figure 2b gives a picture which clarifies the definition, showing a space with 4 connected components that has only 3 effective components. If the optimum $\theta_*$ belongs to the same effective component as the initialization, then it is possible to reach a parameter that is observationally equivalent to it (through permutations and rescalings).

We present a useful result which tells us that the action of rescaling of Equation (3) can take any non-degenerate parameter $\theta \in \mathcal{H}(c)$ to any other invariant set $\mathcal{H}(c')$ for every $c' \in \mathbb{R}^l$. This means that any invariant set can realize all the neural network's functions.

**Proposition 4.** *For every $c_k \in \mathbb{R}$ and for every $\theta_k \in \Theta_k$ such that $W_k^{(1)}, W_k^{(2)} \neq 0$, there exists a unique $\alpha_k \in \mathbb{R}_+$ such that $T_{\alpha_k}(\theta_k) \in \mathcal{Q}(c_k)$. If $W_k^{(1)} = 0$ and $W_k^{(2)} \neq 0$, then the same holds for every $c_k < 0$, while, if $W_k^{(1)} \neq 0$ and $W_k^{(2)} = 0$, it holds for every $c_k > 0$.*

The proof can be found in Appendix D.5 with the formula of the specific $\alpha$ which realizes the rescaling.

The following theorem leverages the power of Proposition 4 to give necessary and sufficient conditions for $\theta$ and $\theta'$ to belong to the same effective component.

**Theorem 2.** *Let $d > 1$ and $e = 1$. Let $c \in \mathbb{R}^l$ and $l_-$ be the number of neurons such that $c_k < 0$. Assume that $l_- \geq 1$. Let $C, C' \subseteq \mathcal{H}(c)$ be two distinct connected components of $\mathcal{H}(c)$ such that $s(\theta) = s$, $\forall \theta \in C$, and $s(\theta') = s'$, $\forall \theta' \in C'$. Then, the following statements are equivalent:*

*1. for every $\theta \in C$ there exists $\theta' \in C'$ such that $\theta \overset{\mathrm{rp}}{\sim} \theta'$;*

*2. $\sum_{i=1}^{l_-} s_i = \sum_{i=1}^{l_-} s_i'$*

The theorem, proven in Appendix D.6, tells us that, while connected components are identified by $s$, the effective components are identified only by the values of $\sum_i s_i$ or, equivalently, by the distribution of $\pm 1$ in $s$. Therefore, we find that the number of effective components scales much slower than the exponential growth of the number of connected components given by Corollary 1.

**Corollary 3.** *The number of effective components of $\mathcal{H}(c)$ is given by $1 + l_-$.*

*Proof.* Theorem 2 tells us that two connected components $C, C'$ belong to the same effective component if and only if their associated sign vectors $s, s' \in \{-1, 1\}^{l_-}$ have the same sum. The number of effective components will thus equal the number of different values that the sum $\sum_{i=1}^{l_-} s_i$ can have. If $s_i = 1 \; \forall i$ then $\sum_{i=1}^{l_-} s = l_-$. Each switch of a component to $-1$ decreases the sum's value by 2 until it reaches the minimum $-l_-$. Therefore, the total number of values of the sum will be $1 + l_-$. $\qquad \square$

## 6 Empirical Validation

**Task, dataset, and model setup.** We display here a toy example, showing how the initialization of the model can cause a topological obstruction, making the optimum unreachable.

We consider the function $F(x_1, x_2) = -(x_1 + x_2)$, which will be our ground-truth. Next, we generate a dataset of 8000 points $(x_i, F(x_i))$ by sampling $x_i \sim U([0, 1]^2)$. Our model, depicted in Figure 3a)

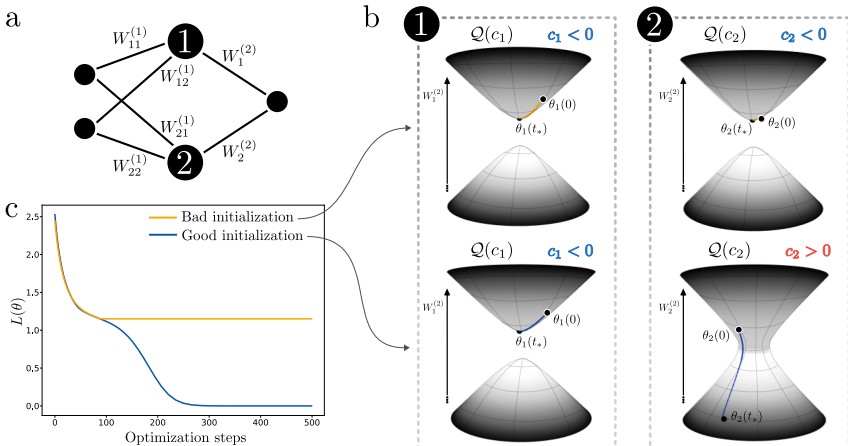

Figure 3: Visualization of the experimental setup described in Section 6. **a.** The small 2-layer neural network architecture considered. **b.** The hidden neurons' parameter spaces, together with the invariant hyperquadrics associated with hidden neurons 1 (left) and 2 (right), for an initialization with topological obstruction (top) and without it (bottom). The colored curves represent the gradient descent trajectories from initialization $\theta_k(0)$ up to $t_* = 500$ optimization steps. **c.** The loss curves for the bad (obstructed) and good initializations.

is a one hidden layer neural network with 2 hidden neurons, ReLU activations and no biases. All the weights are initialized by independently sampling from $U\left(\left[-\sqrt{2}, \sqrt{2}\right]\right)$. From the task and the network's architecture, it is clear that at least one of the output weights has to be negative to approximate $F$ correctly.

To standardize our results, we apply the rescaling of Proposition 4 and relocate the initial parameters to an observationally equivalent one in the invariant set $\mathcal{H}(c)$ with $c_k \in \{-0.1, 0.1\}$, controlling the sign of the weights on the last layer. We allow ourselves to do these two manipulations to control the experiments while only marginally modifying the network initialization, avoiding the introduction of massively unbalanced weights, which could change the dynamics, as shown in Neyshabur et al. [34]. Finally, we train the network using gradient descent on the MSE loss with a small learning rate of $h = 0.01$. This limits the variations of $c_k$ values to less than one percent along training, giving us a good approximation of gradient flow.

**Results.** We initialize different models and collect all states and losses. First, when we initialize the model with an "unlucky" configuration, namely $c = (-0.1, -0.1)$ (the space has 4 connected components) and $s(\theta) = (+1, +1)$, we find that the trajectories are confined to the positive region of their invariant hyperquadric, resulting in a poor approximation of $F$, as we can see in Figure 3b (top) and in the loss of Figure 3c. Instead, with an initial configuration such that $c = (-0.1, +0.1)$ (2 connected components) and $s(\theta) = (+1, +1)$, the model can leverage the connectedness of $\mathcal{Q}(c_2)$ to learn $F$ by flipping the sign of the second neuron's output weight (Figure 3b bottom right).

**A more realistic experiment.** We present here a further experiment to show how the topological obstruction can be a hindrance in a more realistic setting. We consider a simple binary classification task on the well-known *breast cancer* dataset [49], which we try to solve by fitting a one-layer ReLU neural network trained to minimize the BCE loss. We vary the number of hidden neurons $l$ and, for each $l$, we change the number of non-pathological neurons $l_+$ (neurons with $c_k > 0$) from 0 to $l$. We repeat the experiment with 100 different random initializations and show how the model's average performance changes when the degree of disconnectedness of its invariant set is varied. The result, on the left panel of Figure 4, clearly shows the presence of a "gradient" in performance, where increasing the number of non-pathological neurons decreases the average value of the test loss after training. The right panel of Figure 4, moreover, shows how the impact of the obstruction depends on the number of non-pathological neurons and not on their fraction over the total number of hidden neurons.

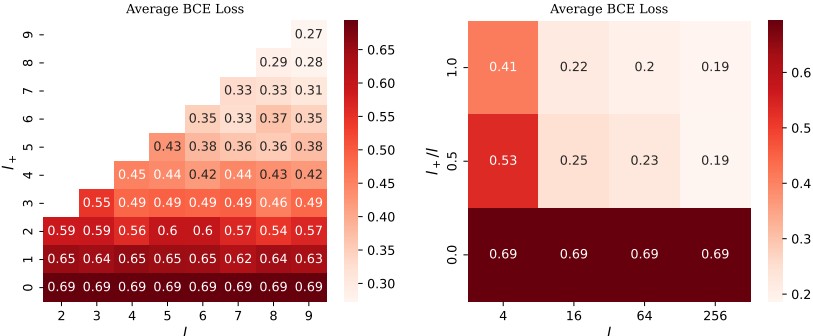

Figure 4: **Left.** Average test BCE loss of a two-layer ReLU neural network trained on the *breast cancer* dataset over 100 different initializations for each pair $(l, l_+)$, $l = 2, \ldots, 9$ and $l_+ \leq l$, of numbers of hidden neurons and non-pathological neurons. **Right.** the y-axis displays the percentage of non-pathological neurons.

# 7 Conclusions

In this paper, we have given analytical results that clarify the nature of the constraints imposed by gradient flow on the parameter space of a two-layer neural network with homogeneous activations. In the case of a single scalar output, which appears in tasks such as binary classification and scalar regression, we identified initial conditions that lead to a topological obstruction in the form of the parameter space's fragmentation into multiple connected components. This is caused by pathological neurons whose output weights cannot change their sign during training. Moreover, if one also considers the network's symmetries under permutations of the hidden neurons, we find that most of the connected components are equivalent. The number of effective components of the resulting space scales linearly with the number of pathological neurons, contrasting with the exponential growth of the number of connected components obtained without considering the permutation symmetries.

As shown in the last numerical experiment, the lack of non-pathological neurons hinders learning, even when the network's width is scaled. Our probabilistic analysis outlined in Appendix F, however, shows that with common initialization schemes, the probability of creating a pathological neuron decreases rapidly with increased inner layer width. Therefore, the combination of specific initialization schemes and a large number of hidden neurons (beyond the minimum required to solve a task) appears to make this obstruction unlikely in practice. This work describes a simple safeguard to avoid obstructions, which can, for instance, discourage the usage of initialization schemes that result in the proliferation of pathological neurons.

# 8 Limitations

The main limitation of the work is the network's architecture, which is limited to only one hidden layer. Considering multiple layers, we can still define rescalings and permutations and find invariant hyperquadrics for each hidden neuron. The issue emerges in the fact that these hyperquadrics are not "independent" anymore, and the invariant set cannot be factored into the product of the $\mathcal{Q}(c_k)$. This intuitively results from the fact that in the multi-layer case, each weight in the hidden layers is shared by two neurons.

The second limitation is that our study focuses on gradient flow optimization. This idealized situation doesn't take into account the fact that moderate step size of gradient descent and stochastic gradient descent can break the conservation of $\langle\!\langle \theta, \theta \rangle\!\rangle_k$ and make the parameters drift away from the invariant set [2]. Moreover, popular optimizers like ADAM [25] update the parameters employing the gradients at previous iterations so their trajectories will not be constrained to lie on $\mathcal{H}(c)$ as we defined it.

The inclusion of regularization terms in the loss function, such as $\ell_p$ regularizations, also breaks the invariance to rescalings.

## Acknowledgements

M.N. acknowledges the project PNRR-NGEU, which has received funding from the MUR – DM 352/2022.

F.V. would like to thank the Isaac Newton Institute for Mathematical Sciences for the support and hospitality during the programme Hypergraphs: Theory and Applications when work on this paper was undertaken. This work was supported by: EPSRC Grant Number EP/V521929/1

This study was carried out within the FAIR - Future Artificial Intelligence Research and received funding from the European Union Next-GenerationEU (PIANO NAZIONALE DI RIPRESA E RESILIENZA (PNRR) – MISSIONE 4 COMPONENTE 2, INVESTIMENTO 1.3 – D.D. 1555 11/10/2022, PE00000013). This manuscript reflects only the authors' views and opinions; neither the European Union nor the European Commission can be considered responsible for them.

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

# A    Parameter spaces

Let $(e_{11}, e_{12}, \ldots, e_{ll})$ and $(e_1, \ldots, e_l)$ be the canonical bases of the spaces $\mathbb{R}^{l \times l}$ and $\mathbb{R}^l$, respectively and

$$\Theta_k = \left\{ (e_{kk} W^{(1)}, W^{(2)} e_{kk}) \mid (W^{(1)}, W^{(2)}) \in \Theta \right\} \subset \Theta.$$

$\Theta_k$, notice, is the subspace of $\Theta$ consisting of the weight matrices $W^{(1)}$ with null rows except for the $k$-th one, and weight matrices $W^{(2)}$ with null columns except for the $k$-th one. We can check that $\Theta = \Theta_1 \oplus \cdots \oplus \Theta_l$, because, if $I_l$ is the $l \times l$ identity matrix, $\sum e_{kk} = I_l$ so that

$$(W^{(1)}, W^{(2)}) = \sum_k (W_k^{(1)}, W_k^{(2)}) = \sum_k (e_{kk} W^{(1)}, W^{(2)} e_{kk}).$$

Moreover, $\Theta \cong \Theta_1 \times \cdots \times \Theta_l$ via the linear isomorphism

$$\theta = (W^{(1)}, W^{(2)}) \leftrightarrow (\theta_k)_{k=1}^l = \left( (e_{kk} W^{(1)}, W^{(2)} e_{kk}) \right)_{k=1}^l.$$

This, we see, is equivalent to decomposing the neural network of Equation (1) into the computations of the single hidden neurons. Indeed, let $f(x; \theta_k) := f(x, (e_{kk} W^{(1)}, W^{(2)} e_{kk}))$, then, considering that $e_{kk} e_{kk} = e_{kk}$ and that $\sigma(e_{kk} v) = e_{kk} \sigma(v)$, it holds that

$$f(x; \theta_k) = W^{(2)} e_{kk} \sigma(W^{(1)} x) \ \forall k = 1, \ldots, l.$$

Therefore $\sum_k f(x; \theta_k) = f(x; \theta)$.

# B    Primer on topology

Here, we recall some basic facts about the topology required to understand the paper's results. A self-consistent introduction is outside this work's scope, so we refer the interested reader to more complete expositions in Munkres [33, 32].

**Topological manifold.** An $n$-dimensional *topological manifold* is a topological space $X$ which locally looks like the Euclidean space $\mathbb{R}^n$. More formally, for each $p \in X$, there exists a neighbourhood $U$ of $p$ and a homeomorphism mapping $U$ to an open subset of $\mathbb{R}^n$.

**Contractible space.** A topological space $X$ is *contractible* if it can continuously deform to a point $p \in X$. This means that there exists a continuous map

$$F : X \times [0, 1] \to X$$

such that $F(x, 0) = x$ and $F(x, 1) = p$ for every $x \in X$.

**Betti numbers.** Betti numbers formalize the notion of the hole in a topological space and extend it to describe higher-dimensional cavities. The general idea is that one can associate a sequence of Abelian groups named *homology groups* to any space $X$, which encodes rich information about the higher-dimensional cavities in $X$. For what we are concerned here, the rank of the $k$-th homology group is called the $k$-th *Betti number* $\beta_k(X)$. $\beta_k(X)$ counts the number of $k$-dimensional holes in the space: $\beta_0(X)$ count the number of connected components, $\beta_1(X)$ the number of "circular" holes and $\beta_2(X)$ the number of voids or cavities.

A contractible space $X$ is connected and cannot have any holes, and thus its Betti numbers are $\beta_0(X) = 1$ and $\beta_i(X) = 0 \ \forall i > 0$.

Betti numbers are *topological invariants*, meaning they are preserved when a space is transformed via a homeomorphism, namely a bijective, continuous map with continuous inverse.

**Poincaré polynomials.** The *Poincaré polynomial* of a topological space $X$ is the polynomial whose $k$-th coefficient is given by the $k$-th Betti number

$$p_X(x) = \beta_0(X) + \beta_1(X)x + \beta_2(X)x^2 + \ldots.$$

**Künneth formula.** Künneth's theorem describes assembling the homology groups of a Cartesian product of spaces $X \times Y$ from the homology groups of the factors $X, Y$. One of its corollaries tells us that if we care only about Betti numbers, a simple relation holds between the Poincaré polynomials of $X \times Y$ and the ones of $X$ and $Y$, namely,

$$p_{X \times Y}(x) = p_X(x)p_Y(x).$$

**Types of connectedness.** In topology, there are several kinds of connectedness. Two of them are particularly important for this work.

**1.)** A topological space $X$ is *connected* if it cannot be divided into two disjoint non-empty open sets. If it is not connected, the connected component of a point $x \in X$ is given by the union of all connected subsets of $X$ which contain $x$.

A topological space equal to the Cartesian product of two spaces $X = Y \times Z$ is connected if and only if $Y$ and $Z$ are both connected.

**2.)** A topological space is *path-connected* if, for every pair of points $x, y \in X$, there exists a continuous curve $\gamma : [0,1] \to X$ such that $\gamma(0) = x$, $\gamma(1) = y$. The *path-component* of $x$ is the set of all $y \in X$ such that a continuous curve exists connecting $x$ to $y$.

This second notion is more relevant to our setting, where we care about the possible destinations of the optimization trajectories.

Path-connectedness implies connectedness, but not the opposite. There are situations, however, where these two notions are equivalent. For example, when $X$ is a topological manifold, $X$ is connected if and only if $X$ is path connected.

Notice that the 0-th Betti number $\beta_0(X)$ counts the number of connected components but, in general, not the number of path components. With Lemma 2, we prove that these two notions are equivalent for our object of study.

## C  Extra propositions and lemmas

**Lemma 2.** *The invariant set $\mathcal{H}(c)$ is connected if and only if it is path connected.*

*Proof.* Lemma 1 tells us that $\mathcal{H}(c) \cong \mathcal{Q}(c_1) \times \cdots \times \mathcal{Q}(c_l)$.

Let us focus on a particular $\mathcal{Q}(c_k)$.

When $c_k \neq 0$, Proposition 2 tells us that $\mathcal{Q}(c_k)$ is a topological manifold, and thus, it is connected if and only if it is path connected.

When $c_k = 0$, $\mathcal{Q}(0)$ is not a topological manifold but contractible, implying that it is connected. Let us prove that it is also path-connected.

Let $\theta_k, \theta'_k \in \mathcal{Q}(0)$ and define the curve $\gamma : [0,1] \to \mathcal{Q}(0)$

$$\gamma_{k;\theta}(t) = t \cdot \theta_k$$

such that $\gamma_{k;\theta}(0) = \theta$, $\gamma_{k;\theta}(1) = 0$. $\gamma_{k;\theta}(t) \in \mathcal{Q}(0)$ for every $t \in [0,1]$ because

$$\langle\!\langle \gamma_{k;\theta}(t), \gamma_{k;\theta}(t) \rangle\!\rangle_k = t \langle\!\langle \theta, \theta \rangle\!\rangle_k = 0.$$

Therefore, the segment from $\theta_k$ to $0$ belongs to $\mathcal{Q}(0)$.

A continuous curve from $\theta_k$ to $\theta'_k$ can be thus obtained by

$$\gamma_{k;\theta'}\gamma_{k;\theta}(t) := \begin{cases} \gamma_{k;\theta}(2t) & \text{if } t \in [0, \frac{1}{2}] \\ \gamma_{k;\theta'}(2 - 2t) & \text{if } t \in [\frac{1}{2}, 1] \end{cases}$$

which is continuous because $\gamma_{k;\theta}(1) = \gamma_{k;\theta'}(1) = 0$. Therefore $\mathcal{Q}(0)$ is path connected.

Finally, if $\mathcal{H}(c)$ is connected, then all of its factors $\mathcal{Q}(c_k)$ are connected, which, in turn, is true if and only if they are path-connected. A product of path-connected space is again path-connected, and therefore $\mathcal{H}(c)$ is path-connected. The other implication is true because path-connectedness implies connectedness, thus concluding the proof. $\square$

**Lemma 3** (Interchange of rescalings and permutations). *Let $\theta \in \Theta$, $\alpha \in \mathbb{R}_+^l$ and $\pi \in \mathfrak{S}_l$, then, if $\tilde{\alpha} = R_{\pi^{-1}}(\alpha)$*

$$T_\alpha P_\pi(\theta) = P_\pi T_{\tilde{\alpha}}(\theta). \tag{16}$$

*Proof.* Given that

$$T_\alpha P_\pi(\theta) = (\mathrm{diag}(\alpha) R_\pi W^{(1)}, W^{(2)} R_\pi^\top \mathrm{diag}(\alpha)^{-1})$$

we need to prove that $\mathrm{diag}(\alpha) R_\pi = R_\pi \mathrm{diag}(\tilde{\alpha})$.

$$(\mathrm{diag}(\alpha) R_\pi)_{ij} = \sum_{k=1}^l \mathrm{diag}(\alpha)_{ik} (R_\pi)_{kj} = \alpha_i (R_\pi)_{ij} = \begin{cases} \alpha_i & \text{if } j = \pi(i) \\ 0 & \text{otherwise} \end{cases}.$$

Let us pick a generic $\tilde{\alpha} \in \mathbb{R}_+^l$.

$$(R_\pi \mathrm{diag}(\tilde{\alpha}))_{ij} = \sum_{k=1}^l (R_\pi)_{ik} \mathrm{diag}(\tilde{\alpha})_{kj} = (R_\pi)_{ij} \tilde{\alpha}_j = \begin{cases} \tilde{\alpha}_j & \text{if } j = \pi(i) \\ 0 & \text{otherwise} \end{cases}.$$

Let us consider the inverse permutation $\pi^{-1}$ so that $\pi^{-1}(j) = i$ if $\pi(i) = j$. Then, if $\tilde{\alpha} = R_{\pi^{-1}}\alpha$,

$$\tilde{\alpha}_j = \alpha_{\pi^{-1}(j)} = \alpha_i$$

and thus we get that $\mathrm{diag}(\alpha) R_\pi = R_\pi \mathrm{diag}(\tilde{\alpha})$. $\qquad\square$

## D  Proofs

### D.1  Proof of Lemma 1

*Proof.* Proposition 1 tells us that the invariant set can be decomposed as the direct sum of the single hidden neurons' parameter spaces. This means that, for every $\theta \in \Theta$, there exist unique $\theta_1 \in \Theta_1, \ldots, \theta_l \in \Theta_l$ such that

$$\theta = \theta_1 + \theta_2 + \cdots + \theta_l.$$

Therefore, we have a linear isomorphism $\varphi : \Theta_1 \times \cdots \times \Theta_k \to \Theta$

$$\varphi : (\theta_1, \ldots, \theta_l) \mapsto \theta_1 + \cdots + \theta_l = \theta.$$

The invariant set is a subset of $\Theta$, which is given as the set of solutions of $l$ equations $\langle\!\langle \theta, \theta \rangle\!\rangle_k = c_k$ $k = 1, \ldots, l$. Notice that each of these equations involves a set of variables that appear only in that particular equation. These variables are exactly the ones which belong to $\Theta_k$. In fact $\langle\!\langle \theta, \theta \rangle\!\rangle_k = \langle\!\langle \theta_k, \theta_k \rangle\!\rangle_k$.

Therefore, given $\theta_1 \in \mathcal{Q}(c_1), \ldots, \theta_l \in \mathcal{Q}(c_l)$ we have that

$$\varphi(\theta_1, \ldots, \theta_l) = \sum_{k=1}^l \theta_k \in \mathcal{H}(c).$$

On the opposite, given $\theta \in \mathcal{H}(c)$ we have that

$$\varphi^{-1}(\theta) = (\theta_1, \ldots, \theta_l) \in \mathcal{Q}(c_1) \times \cdots \times \mathcal{Q}(c_l).$$

Therefore, $\mathcal{H}(c)$ is in bijection with $\mathcal{Q}(c_1) \times \cdots \times \mathcal{Q}(c_l)$ through $\varphi$ which, being a linear isomorphism, implies also that $\mathcal{H}(c)$ and $\mathcal{Q}(c_1) \times \cdots \times \mathcal{Q}(c_l)$ are homeomorphic. $\qquad\square$

### D.2  Proof of Proposition 2

*Proof.* Let us consider the three cases separately. If $c_k > 0$, $\mathcal{Q}(c_k)$ is defined by the equation

$$\sum_{i=1}^d \left(W_{ki}^{(1)}\right)^2 - \sum_{j=1}^e \left(W_{jk}^{(2)}\right)^2 = c_k \iff \left\|W_k^{(1)}\right\|_F^2 - \left\|W_k^{(2)}\right\|_F^2 = c_k,$$

where $\|\cdot\|_F$ is the Frobenius norm of a matrix, i.e., the square root of the sum of the squares of its elements. This can be rewritten as

$$\left\|W_k^{(1)}\right\|_F = \sqrt{c_k + \left\|W_k^{(2)}\right\|_F^2} \tag{17}$$

where $\sqrt{c_k + \left\|W_k^{(2)}\right\|_F^2} > 0$ because $c_k > 0$. We define the map $h : \mathcal{Q}(c_k) \to S^{d-1} \times \mathbb{R}^e$ as

$$h\left(W_k^{(1)}, W_k^{(2)}\right) = \left(\frac{W_k^{(1)}}{\sqrt{c_k + \left\|W_k^{(2)}\right\|_F^2}}, W_k^{(2)}\right) \tag{18}$$

where, notice, the first component belongs to the sphere $S^{d-1}$ because of Equation (17) and $W_k^{(2)} \in \mathbb{R}^e$. This map is bijective, differentiable and has the following inverse $h^{-1} : S^{d-1} \times \mathbb{R}^e \to \mathcal{Q}(_k)$

$$h^{-1}(u, x) = (\sqrt{c_k + \|v\|_F^2}\, u, x)$$

which is differentiable. Therefore, $h$ is a diffeomorphism from $\mathcal{Q}(c_k)$ to $S^{d-1} \times \mathbb{R}^e$.

If $c_k < 0$, we write the equation of $\mathcal{Q}(c_k)$ as

$$\left\|W_k^{(2)}\right\|_2 = \sqrt{-c_k + \left\|W_k^{(1)}\right\|_2^2}$$

where $W_k^{(1)}$ and $W_k^{(2)}$ have switched their role to guarantee the term on the right to be positive. The diffeomorphism is now built analogously to Equation (18) as a map $h : \mathcal{Q}(c_k) \to S^{e-1} \times \mathbb{R}^d$.

If $c_k = 0$, we prove that $\mathcal{Q}(0)$ is a contractible space. To do that, we exhibit a homotopy equivalence between $\mathcal{Q}(0)$ and the point 0, i.e. a continuous map $p : [0,1] \times \mathcal{Q}(0) \to \mathcal{Q}(0)$ such that $p(0, \theta_k) = \theta_k$ and $p(1, \theta_k) = 0 \; \forall \theta_k \in \mathcal{Q}(0)$. The map is defined in the following way:

$$p(\lambda, \theta_k) = (1 - \lambda)\theta_k.$$

This is continuous and well-defined because

$$\langle\!\langle p(\lambda, \theta_k), p(\lambda, \theta_k)\rangle\!\rangle_k = \langle\!\langle(1-\lambda)\theta_k, (1-\lambda)\theta_k\rangle\!\rangle_k = (1-\lambda)^2\langle\!\langle\theta_k, \theta_k\rangle\!\rangle_k = 0,$$

meaning that $p(\lambda, \theta_k) \in \mathcal{Q}(0)$ for every $\theta_k \in \mathcal{Q}(0)$ and for every $\lambda \in [0,1]$. $\qquad \square$

### D.3 Proof of Theorem 1

*Proof.* An implication of the Künneth formula is that the Poincaré polynomial of the Cartesian product of two spaces is equal to the product of their Poincaré polynomials:

$$p_{X \times Y}(x) = p_X(x)p_Y(y).$$

Starting from Proposition 2, we can apply this result to $\mathcal{Q}(c_k)$.

$$p_{\mathcal{Q}(c_k)} = \begin{cases} p_{\mathbb{R}^e}(x)p_{S^{d-1}}(x) & \text{if } c_k > 0 \\ p_{\mathbb{R}^d}(x)p_{S^{e-1}}(x) & \text{if } c_k < 0 \\ 1 & \text{if } c_k = 0 \end{cases} \tag{19}$$

because a contractible space has 1 connected component and all of its other Betti numbers equal to zero.

Moreover, we know that $\mathbb{R}^n$ is contractible for any $n$ and its Poincaré polynomial is $p_{\mathbb{R}^n}(x) = 1$. The Poincaré polynomial of the sphere $S^n$ is given by $p_{S^n}(x) = 1 + x^n$.

Equation (19) becomes

$$p_{\mathcal{Q}(c_k)} = \begin{cases} 1 + x^{d-1} & \text{if } c_k > 0 \\ 1 + x^{e-1} & \text{if } c_k < 0 \\ 1 & \text{if } c_k = 0 \end{cases} . \tag{20}$$

Given that Lemma 1 tells us that $\mathcal{H}(c)$ can be factored into the product of the $\mathcal{Q}(c_k)$, we apply Künneth formula and find that

$$p_{\mathcal{H}(c)}(x) = p_{\mathcal{Q}(c_1)}(x) \cdots p_{\mathcal{Q}(c_l)}(x) = (1 + x^{d-1})^{l_+}(1 + x^{e-1})^{l_-}. \tag{21}$$

$$\square$$

## D.4 Proof of Proposition 3

*Proof.* In the following, we exploit Lemma 2 and use the terminology connected and path-connected interchangeably.

Let us first prove that $s(\theta) = s(\theta')$ means that $\theta$ and $\theta'$ belong to the same connected component. We do this by explicitly building a continuous curve $\delta : [0, 1] \to \mathcal{H}(c)$ such that $\delta(0) = \theta$ and $\delta(1) = \theta'$.

Let us proceed by leveraging the homeomorphism from $\mathcal{H}(c)$ and $\mathcal{Q}(c_1) \times \cdots \times \mathcal{Q}(c_l)$ and consider the different components of $\delta$ in the invariant hyperquadrics associated to each neuron $\delta = (\delta_1, \ldots, \delta_l)$.

If $c_k \geq 0$, we know that $\mathcal{Q}(c_k)$ is path-connected and therefore we fix $\delta_k(t)$ to any continuous curve in $\mathcal{Q}(c_k)$ such that $\delta_k(0) = \theta_k$ and $\delta_k(1) = \theta'_k$.

If $c_k < 0$ for $k \in K := \{k_1, \ldots, k_{l_-}\} \subseteq \{1, \ldots, l\}$, we define the curve $\gamma_{i;\theta} : [0, 1] \to \mathcal{Q}(c_{k_i})$ with

$$\gamma_{i;\theta}(t) = \left( (1-t)W^{(1)}_{k_i 1}, \ldots, (1-t)W^{(1)}_{k_i l}, s(\theta)_i \sqrt{-c_{k_i} + (1-t)^2 \left\| W^{(1)}_{k_i} \right\|_F^2} \right)$$

for every $i = 1, \ldots, l_-$.

$\gamma_{i;\theta}$ is a continuous curve which connects the point $\gamma_{i;\theta}(0) = \theta_{k_i}$ with $\gamma_{i;\theta}(1) = (0, s(\theta)_i \sqrt{-c_{k_i}})$.

If we define $\bar{\gamma}_{i;\theta}(t) := \gamma_{i;\theta}(1 - t)$, which is the same curve as $\gamma_{i;\theta}$ but traversed in the opposite direction, we can define the curve

$$\delta_{k_i}(t) = \bar{\gamma}_{i;\theta'} \gamma_{i;\theta}(t)$$

i.e. the curve which travels on $\gamma_{i;\theta}$ for $t \in [0, \frac{1}{2}]$ and on $\bar{\gamma}_{i;\theta'}$ for $t \in [\frac{1}{2}, 1]$, for $i = 1, \ldots, l_-$.

Notice now that $\delta_{k_i}(0) = \theta_{k_i}$ and $\delta_{k_i}(1) = \theta'_{k_i}$. Moreover $\delta_{k_i}$ is continuous because

$$\gamma_{i;\theta}(1) = (0, s(\theta)_i \sqrt{-c_{k_i}}) = (0, s(\theta')_i \sqrt{-c_{k_i}}) = \bar{\gamma}_{i;\theta'}(0)$$

under the hypothesis that $s(\theta) = s(\theta')$.

Finally, we found a continuous curve $\delta = (\delta_1, \ldots, \delta_l)$ such that $\delta(t) \in \mathcal{H}(c) \ \forall t \in [0, 1]$ and $\delta(0) = \theta$, $\delta(1) = \theta'$. Therefore, $\theta$ and $\theta'$ belong to the same connected component.

Let us now prove that if $\theta$ and $\theta'$ belong to the same connected component, then $s(\theta) = s(\theta')$.

Let $\gamma : [0, 1] \to \mathcal{H}(c)$ be a continuous curve in $\mathcal{H}(c)$ such that $\gamma(0) = \theta$ and $\gamma(1) = \theta'$.

For each $k \in K = \{k_1, \ldots, k_{l_-}\}$ such that $c_k < 0$, we know that $\gamma_k(t) \in \mathcal{Q}(c_k)$ means that

$$\gamma_k(t) = \Big( \gamma^{(1)}_{k1}(t), \ldots, \gamma^{(1)}_{kd}(t), \underbrace{s_k(t)\sqrt{-c_k + \left\| \gamma^{(1)}_k \right\|_F^2}}_{\gamma^{(2)}_k(t)} \Big)$$

for some function $s_k(t) \in \{-1, 1\}$ such that $s_k(0) = s(\theta)_k$ and $s_k(1) = s(\theta')_k$.

Assume, by contradiction, that $s(\theta)_k = -s(\theta')_k$. Assume also that $s(\theta)_k = +1$ and $s(\theta')_k = -1$.

Notice that $c_k < 0$ implies that $p(t) := \sqrt{-c_k + \left\| \gamma^{(1)}_k \right\|_F^2} > 0$.

The function $\gamma^{(2)}_k(t) = s_k(t)p(t)$, then, is a continuous function such that $\gamma^{(2)}_k(0) > 0$ and $\gamma^{(2)}_k(1) < 0$ and thus, by the intermediate value theorem, there exists $t_* \in (0, 1)$ such that $\gamma^{(2)}_k(t_*) = 0$.

But $\gamma^{(2)}_k(t) \neq 0$ for every $t$, as $s_k(t) \in \{-1, 1\}$ and $\sqrt{-c_k + \left\| \gamma^{(1)}_k \right\|_F^2} > 0$.

Repeating the argument for $s(\theta)_k = -1$ we then prove by contradiction that $s(\theta)_k = s(\theta')_k$ for all $k \in K$, thus concluding the proof. $\square$

## D.5 Proof of Proposition 4

*Proof.* We have by Equation (10) and Equation (2):

$$\langle\!\langle T_\alpha(\theta), T_\alpha(\theta) \rangle\!\rangle_k - c_k = 0 \iff \alpha_k^2 \sum_{i=1}^{d} (W^{(1)}_{ki})^2 - \frac{1}{\alpha_k^2} \sum_{j=1}^{e} (W^{(2)}_{jk})^2 - c_k = 0$$

By renaming $A = \sum_{i=1}^{d} (W_{ki}^{(1)})^2 = \left\| W_k^{(1)} \right\|_F^2$, $C = \sum_{j=1}^{e} (W_{jk}^{(2)})^2 = \left\| W_k^{(2)} \right\|_F^2$ and multiplying by $\alpha_k^2 > 0$ we have:

$$A\alpha_k^4 - c_k \alpha_k^2 - C = 0 \tag{22}$$

Solving for $\alpha_k^2$ gives us:

$$\Delta = c_k^2 + 4AC \geq 4AC > 0$$

$$\alpha_k^2 = \frac{c_k \pm \sqrt{\Delta}}{2A}$$

Given that we want $\alpha_k > 0$, we discard the negative solution. The other is positive because $\Delta > c_k^2$ and thus $\sqrt{\Delta} > |c_k|$.

$$\alpha_k = \pm \sqrt{\frac{c_k + \sqrt{\Delta}}{2A}}$$

Of which we keep the positive solution only, with its full expression being:

$$\alpha_k = \sqrt{\frac{c_k + \sqrt{c_k^2 + 4 \left\| W_k^{(1)} \right\|_F^2 \left\| W_k^{(2)} \right\|_F^2}}{2 \left\| W_k^{(1)} \right\|_F^2}}. \tag{23}$$

Hence, if $\alpha = (\alpha_1, \ldots, \alpha_l)$ with $\alpha_k$ given by Equation (23), we get that $\langle\!\langle T_\alpha(\theta), T_\alpha(\theta) \rangle\!\rangle_k = c_k \ \forall k = 1, \ldots, l$.

Let us consider now the pathological cases $W_k^{(1)} = 0$ or $W_k^{(2)} = 0$.

If $W_k^{(1)} = 0, W_k^{(2)} \neq 0$ then $A = 0, C \neq 0$. Therefore, we have that Equation (22) becomes

$$-c_k \alpha_k^2 - C = 0$$

which has solutions if and only if $c_k < 0$. In that case $\alpha_k = \frac{\left\| W_k^{(2)} \right\|_F}{\sqrt{-c_k}}$. This means that a hidden neuron with zero input weights and nonzero output weights can be rescaled only to the invariant hyperquadrics with $c_k < 0$.

If $W_k^{(2)} = 0, W_k^{(1)} \neq 0$ then $C = 0, A \neq 0$. Therefore, we have that Equation (22) becomes

$$\alpha_k^2 A - c_k = 0$$

which has solutions if and only if $c_k > 0$. In that case $\alpha_k = \frac{\sqrt{c_k}}{\left\| W_k^{(1)} \right\|_F}$. This means that a hidden neuron with zero output weights and nonzero input weights can be rescaled only to the invariant hyperquadrics with $c_k > 0$.

If $W_k^{(1)} = 0$ and $W_k^{(2)} = 0$, then $\theta_k = 0 \in \mathcal{Q}(0)$ and it cannot be rescaled to any other invariant hyperquadric.

$\square$

### D.6 Proof of Theorem 2

*Proof.* Let us first prove that, if for every $\theta \in C$ there exists a $\theta' \in C'$ such that $\theta \overset{\text{rp}}{\sim} \theta'$ then $\sum_{i=1}^{l_-} s(\theta)_i = \sum_{i=1}^{l_-} s(\theta')_i$.

First, Lemma 3 tells us that we can interchange rescaling and permutation if we permute the rescaling factors accordingly. This means we can reduce any composite action of rescalings and permutations to the action of a single rescaling and a single permutation.

Let $\theta \in C$ and $\theta' \in C'$ such that $\theta \overset{\text{rp}}{\sim} \theta'$. Then there exist $\alpha \in \mathbb{R}_+^l$ and $\pi \in \mathfrak{S}_l$ such that

$$T_\alpha P_\pi(\theta) = \theta'. \tag{24}$$

This means that $T_\alpha P_\pi(\theta)$ and $\theta'$ belong to the same invariant set and, specifically, to the same connected component. Therefore,

$$s(T_\alpha P_\pi(\theta)) = s(\theta').$$

Notice that

$$s(T_\alpha P_\pi(\theta)) = s(P_\pi(\theta))$$

because $T_\alpha$ does not change the sign of $W^{(2)}$ as it acts by scaling it by positive factors. Let us focus on

$$s(P_\pi(\theta)) = \text{sign}((W^{(2)} R_\pi^\top)_-).$$

If the neurons of $\theta$ such that $c_k < 0$ are indexed by $k_1, k_2, \ldots, k_{l_-}$, we will have that the neurons of $P_\pi(\theta)$ such that $c_k < 0$ are indexed by $\pi(k_1), \pi(k_2), \ldots, \pi(k_{l_-})$. Therefore

$$s(P_\pi(\theta))_i = \text{sign}(W^{(2)}_{\pi(k_i)}) = (s(\theta) R_{\pi_-}^\top)_i$$

for some permutation $\pi_- \in \mathfrak{S}_{l_-}$. Therefore, Equation (24) implies that

$$s(\theta') = s(P_\pi(\theta)) = s(\theta) R_{\pi_-}^\top.$$

The action of rescaling and permutation can only reshuffle the label $s$ of the connected component. This means that

$$s(\theta) R_{\pi_-}^\top = s(\theta') \implies \sum_{i=1}^{l_-} (s(\theta) R_{\pi_-}^\top)_i = \sum_{i=1}^{l_-} s(\theta')_i \implies \sum_{i=1}^{l_-} s(\theta)_i = \sum_{i=1}^{l_-} s(\theta')_i.$$

Let us now prove the other implication. Let $s, s' \in \mathbb{R}^{l_-}$ such that $\sum_{i=1}^{l_-} s_i = \sum_{i=1}^{l_-} s'_i$.

Given that their sum is equal, $s$ and $s'$ have the same number of $+1$ and $-1$ and thus there exists a permutation $\pi_- \in \mathfrak{S}_{l_-}$ such that $s' = s R_{\pi_-}^\top$.

Let $\pi \in \mathfrak{S}_l$ be the permutation which permutes the neurons such that $c_k < 0$ according to $\pi_-$ and leaves the others fixed. In this way $s(P_\pi(\theta)) = s R_{\pi_-}^\top = s'$.

$P_\pi(\theta)$, however, doesn't belong to $\mathcal{H}(c)$ but to another invariant set given by $\mathcal{H}(R_\pi c)$.

Applying Proposition 4 we can find a rescaling $\alpha = \alpha(\pi) \in \mathbb{R}_+^l$ such that $T_\alpha P_\pi(\theta) \in \mathcal{H}(c)$.

Since neurons such that $c_k \geq 0$, are left unchanged by the permutation, we can apply the proposition and we rescale them with $\alpha_k = 1$. The permuted neurons are the ones such that $c_k < 0$, namely the ones whose weights satisfy $\left\| W_k^{(1)} \right\|_F^2 - \left\| W_k^{(2)} \right\|_F^2 < 0$, meaning that $W_k^{(2)} \neq 0$.

As noted above, the action of the rescaling doesn't change the sign vector, and thus

$$s(T_\alpha P_\pi(\theta)) = s(P_\pi(\theta)) = s'.$$

If we name $\theta' := T_\alpha P_\pi(\theta)$ this result means that we found a $\theta' \overset{\text{rp}}{\sim} \theta$ such that $\theta' \in C'$, thus concluding the proof. $\qquad\square$

## E   Including biases

Let us consider the case where we include biases. The resulting two-layer neural network can be written as

$$f(x; \theta) = W^{(2)} \sigma(W^{(1)} x + b^{(1)}) + b^{(2)}, \tag{25}$$

where $b^{(1)} \in \mathbb{R}^l$ and $b^{(2)} \in \mathbb{R}^e$.

To work with this extended set of parameters, we re-define the space

$$\Theta = \left\{ \theta = (W^{(1)}, b^{(1)}, W^{(2)}, b^{(2)}) \right\}$$

and the single hidden neuron spaces

$$\Theta_k = \left\{\theta_k = \left(e_{kk}W_k^{(1)}, e_{kk}b_k^{(1)}, W_k^{(2)}e_{kk}\right)\right\}$$

where the second bias term $b^{(2)}$ does not appear because it is not directly involved with the computations of the hidden neurons. This means that we can write

$$\Theta \cong \Theta_1 \times \cdots \times \Theta_l \times \mathbb{R}^e$$

where $\mathbb{R}^e$ is included to describe the parameters in $b^{(2)}$.

The neuron rescaling action now acts on the biases $b^{(1)}$ as well as the weights:

$$
\begin{aligned}
T &: \mathbb{R}_+ \times \Theta_k \to \Theta_k \\
(\alpha, \theta_k) &\mapsto T_\alpha(\theta_k) = \left(\alpha W_k^{(1)}, \alpha b_k^{(1)}, \frac{1}{\alpha}W_k^{(2)}\right)
\end{aligned}
\tag{26}
$$

and can be extended to the whole space of parameters

$$
\begin{aligned}
T: \quad \mathbb{R}_+^l \times \Theta \quad &\to \quad \Theta \\
(\alpha, \theta) \quad &\mapsto \quad T_\alpha(\theta) = \left(\mathrm{diag}(\alpha)W^{(1)}, \mathrm{diag}(\alpha)b^{(1)}, W^{(2)}\mathrm{diag}(\alpha)^{-1}, b^{(2)}\right).
\end{aligned}
\tag{27}
$$

Once again, we find that $T_\alpha\theta \sim \theta$.

In this more general case, we can rewrite the bilinear form to include the biases. If $\theta = (W^{(1)}, b^{(1)}, W^{(2)}, b^{(2)})$ and $\eta = (V^{(1)}, p^{(1)}, V^{(2)}, p^{(2)})$, we define

$$\langle\!\langle \theta, \eta \rangle\!\rangle_k = \sum_{i=1}^d W_{ki}^{(1)}V_{ki}^{(1)} + b_k^{(1)}p_k^{(1)} - \sum_{j=1}^e W_{jk}^{(2)}V_{jk}^{(2)} \tag{28}$$

and see that, once gradient flow optimization, we have a conservation condition like the one of Equation (9)

$$\langle\!\langle \theta(t), \theta(t) \rangle\!\rangle_k = c_k \,\forall t > 0 \,\forall k = 1, \ldots, l.$$

Once again, we call $\mathcal{Q}(c_k)$ the hypersurface of $\Theta_k$ which satisfies the equation $\langle\!\langle \theta, \theta \rangle\!\rangle_k = c_k$ and $\mathcal{H}(c_k)$ the set in $\Theta$ defined by $\langle\!\langle \theta, \theta \rangle\!\rangle_k = c_k \,\forall k = 1, \ldots, l$.

With this in mind, it is not hard to extend the results of Proposition 2 and Theorem 1 which turn out to be slightly modified.

**Proposition 5.** *If $c_k > 0$, $\mathcal{Q}(c_k)$ is a topological manifold homeomorphic to $\mathbb{R}^e \times S^d$. If $c_k < 0$, $\mathcal{Q}(c_k)$ is a topological manifold homeomorphic to $\mathbb{R}^d \times S^{e-1}$. If $c_k = 0$, $\mathcal{Q}(0)$ is contractible.*

In this case, we can factor the space of parameters $\mathcal{H}(c)$ as

$$\mathcal{H}(c) \cong \mathcal{Q}(c_1) \times \cdots \times \mathcal{Q}(c_l) \times \mathbb{R}^e,$$

where the last factor is due to the freedom in choosing the values of $b^{(2)}$.

**Proposition 6.** *Let $c_k \neq 0 \,\forall k = 1, \ldots, l$. Let $l_+, l_-, l_0$ be the number of positive, negative and zero elements of c, respectively. The Poincaré polynomial of $\mathcal{H}(c)$ is given by*

$$p_{\mathcal{H}(c)}(x) = (1 + x^d)^{l_+}(1 + x^{e-1})^{l_-} \tag{29}$$

**Corollary 4.** *The $0$-th Betti number $\beta_0(c)$ of $\mathcal{H}(c)$, corresponding to the number of its connected components, is given by*

$$\beta_0(c) = \begin{cases} 1 & \text{if } e > 1 \\ 2^{l_-} & \text{if } e = 1 \end{cases} \tag{30}$$

The result we obtain is similar to Corollary 1 although slightly modified by the fact that having a single input neuron does not cause $\mathcal{H}(c)$ to become disconnected anymore. In the case of $e = 1$, therefore, the picture presented in the main text is left unchanged.

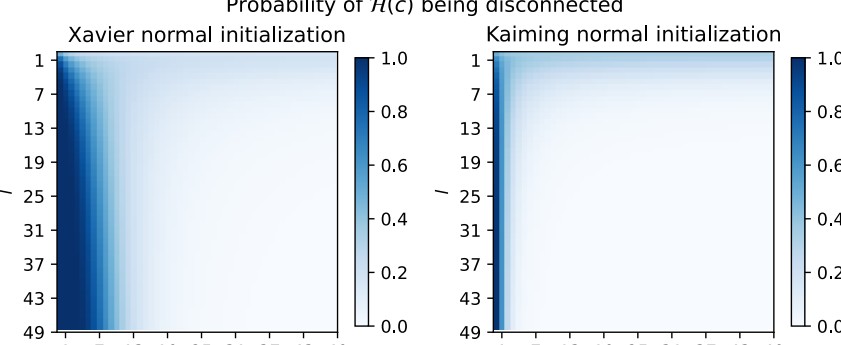

Figure 5: Probability of the topological obstruction as a function of the number of input $d$ and hidden $l$ neurons, when the initial weights are sampled with Xavier normal (left) and Kaiming normal (right) initialization schemes.

## F    Probability of obstruction

Let us consider the following question: *what is the probability of having a disconnected invariant set given a realistic initialization?*

Consider a one-layer ReLU neural network with $e = 1$ and assume that the weights are sampled independently of one another from a normal distribution $W_{ki}^{(1)} \sim \mathcal{N}(0, \sigma_1^2) \; \forall k, i$ , $W_k^{(2)} \sim \mathcal{N}(0, \sigma_2^2) \; \forall k, j$. From Corollary 2 we know that the invariant set $\mathcal{H}(c)$ will be disconnected if and only if there exists a hidden neuron satisfying $\sum_{i=1}^{d}(W_{ki}^{(1)})^2 < (W_k^{(2)})^2$. Given independence of the initial weight sampling, this probability can be computed as

$$\mathbb{P}[\text{obstruction}] = 1 - \mathbb{P}[\sum_{i=1}^{d}(W_{ki}^{(1)})^2 > (W_k^{(2)})^2]^l = 1 - (F_{1,d}(d\sigma_1^2/\sigma_2^2))^l,$$

where $F$ is the cumulative distribution function of the Fisher-Snedecor distribution.

Having obtained this general expression, we can specify it to two common initialization schemes.

- We obtain *Kaiming initialization* [21] with $\sigma_1^2 = 2/d, \sigma_2^2 = 2/l$ resulting in $\mathbb{P}[\text{obstruction}] = 1 - F_{1,d}(l)^l$.
- We obtain *Xavier normal initialization* [19] with $\sigma_1^2 = 2/(d + l), \sigma_2^2 = 2/(1 + l)$ resulting in $\mathbb{P}[\text{obstruction}] = 1 - F_{1,d}(\frac{d+ld}{d+l})^l$.

We plot these two expressions in Figure 5. We can see how, for large values of $d$, the probability of obstruction quickly falls to 0 for any number of hidden neurons. Instead, we see an opposite trend for small values of $d$: the probability of disconnectedness grows with $l$. Moreover, it is interesting to notice that the region of high obstruction probability is much larger for Xavier initialization than for Kaiming initialization, further showing why the latter is preferred when working with ReLU networks.

## G    Intuition on the occurrence of obstruction

We can give some intuition on why there is no obstruction for multiple outputs. First, we consider a single hidden neuron $k$, with $d$ incoming weights and a single output $e = 1$. If the neuron is pathological, we have that

$$\sum_{i=1}^{d}\left(W_{ki}^{(1)}\right)^2 < \left(W_{1k}^{(2)}\right)^2.$$

Since the weights $W_{ij}^{(a)}(t)$ are continuous curves in time, for $W_{1k}^{(2)}$ to change sign, its value needs to pass through 0 but, under the condition above, this cannot happen its square is always positive.

Consider now multiple outputs $e > 1$, resulting in the conservation condition being

$$\sum_{i=1}^{d} \left( W_{ki}^{(1)} \right)^2 < \sum_{j=1}^{e} \left( W_{jk}^{(2)} \right)^2.$$

Now, any component $W_{jk}^{(2)}$ can change sign by passing through 0 because the other components can compensate for it by increasing their magnitude to keep the condition satisfied.

