# OpenReview forum: "Topological obstruction to the training of shallow ReLU neural networks"
_NeurIPS.cc/2024/Conference — NeurIPS 2024 poster_

### Official Review · Reviewer_Gc1k · 2024-07-03

**Soundness:** 3
**Presentation:** 3
**Contribution:** 2
**Rating:** 6
**Confidence:** 4

**Summary:**

This works studies how topological properties affect loss landscape of neural networks, showing that the loss can be divided into disconnected regions where reaching one region from the other using GF is impossible. The theory is shown to have consequence for understanding the training of simple networks

**Strengths:**

Studying topological properties of the loss landscape is interesting and should be encouraged, the theory is solid and convincing

**Weaknesses:**

This paper has quite a few weaknesses in its current form, and these prevent me from recommending acceptance at this stage

1. Prior works. (a) it is unclear how different this work is from Safran et al. [35]. This point is never discussed, and this makes it difficult for me to judge the novelty of this work. (b) Incorrect reference. Eq. (11) appears the earliest in https://arxiv.org/abs/1312.6120, not in Du et al. [11]. (c) emergence of topological obstruction due to permutation/rescaling symmetries have been studied in https://arxiv.org/abs/2309.16932, and the authors need to compare the results to clarify what is novel in the present manuscript

2. The main result applies only to two layer bias-less ReLU nets, which I feel to be too weak

3. The meaningful implications (corollary 1) is only relevant when either the input or the output dimension is 1, which makes the theory very unlikely to the relevant to practice

4. The experiment also has very limited scope. If the authors can show that the effect they studied is relevant for training a much larger model on a more realistic dataset, I would be more convinced of the relevance and contribution of the theory

**Questions:**

Does the results still hold when there is a bias term？

**Limitations:**

The paper could also benefit by discussing more of its limitations

---

> ### Author Rebuttal · Authors · 2024-08-06
>
> Dear Colleague, thank you very much for your useful and insightful review.
> ___
> **Weaknesses.**
>
> **W1.**
> - **(a)** Thanks for this question, an answer to which we believe can improve the quality of our paper. There are several differences, namely: in [35], the input-output layers are restricted to be 1-dimensional; second, they are interested in analyzing the number of PL components for the trained networks, i.e., the decomposition of the Input Space into convex polytopes over which the function associated with the NN is linear. We are rather trying to understand the geometry and the topology of the weights space under GF training. We will insert this clarification in the main text.
> - **(b)** A result close to Eq. (11) is indeed found there, although applied to *linear* neural networks without ReLU activations. We nonetheless agree that it should be mentioned in the main text as preceding Du et al. [11].
> - **(c)** We thank you for highlighting this work, which we were not aware of at the time we submitted ours. Nevertheless, following your pointer, we noticed that it has now been published in ICML. In that paper, the authors study general mirror-reflect symmetries (which include rescaling) of the loss function and derive the constraints that they impose on the gradient and, as a consequence, on gradient-based optimization. While they derive interesting results from their framework, like the emergence of sparsity, their work doesn't deal with the topology of such constraints, which is the main goal of our paper. We will be happy to include it in the references and point out what we have just observed.
>
> **W2.**
>
> As we mention in line 91 of the paper, we discuss the inclusion of biases in Appendix E. The results we find are mostly left unchanged as biases can be treated in the same way as the parameters of the input weights, effectively resulting in a network with $d+1$ inputs.
> This shallow network as a model has limitations but our work could become a theoretical stepstone for studying other architectures. Indeed, this framework may be found inside other models (CNNs) or mechanisms (Graph Attention Network, need to check). Besides, there is an overly extended scientific literature on shallow ReLU networks, as can be appreciated in ref. A B C, and, in particular, in the find survey on shallow with a lot of citations, which will be added to our reference also in answering your important question.
>
> **W3.**
>
> Corollary 1 is a mathematical fact independent of the data nature, and its meaning relies on being a structural characteristic of shallow ReLU tout-court. Concerning the likelihood of facing these situations, there are two distinct considerations to be done: first, there are multiple widely used tasks where the output is a single scalar, for example, binary classification and scalar regression, and there is boundless literature on this architecture; on the other hand, we agree the input dimension equal to 1 may be of theoretical interest only.
>
> **W4.**
>
> Please find in part 2 of the general rebuttal the results of a further experiment that displays the effect of obstruction in a more realistic setting.

---

> > ### Comment · Reviewer_Gc1k · 2024-08-09
> > **Thanks for the detailed reply**
> >
> > The reply addresses my main concerns. Although I still think the main problem is with its unclear relevance. I will raise the contribution rating to fair and the score to 6

---

### Official Review · Reviewer_9QVd · 2024-07-11

**Soundness:** 3
**Presentation:** 3
**Contribution:** 2
**Rating:** 6
**Confidence:** 4

**Summary:**

This paper considers the landscape topology of one hidden layer networks with non-negative homogeneous activations. The main result of this work is that in some cases, the loss landscape may consist of disconnected components that cannot be traversed by gradient flow dynamics. This leads to an obstruction, in the sense that certain initial conditions will result in closed paths away from the global minimum. They further count the number of effective components (in the sense of being equivalent under the symmetries of the problem) and show that this number grows linearly with the number of hidden neurons that cannot change their sign during the dynamics.
The authors provide a full analytical treatment of the problem, supplemented by a simple toy example.

**Strengths:**

I believe the paper has several strengths, in particular:

- **Soundness and clarity** - The authors review some known results, and build upon them in an instructive way, that allows the reader to understand precisely what has been done. Additionally, they provide proofs, as well as intuition for all of their results.

- **Novelty** - As far as I know, this topological analysis is new, with the closest work being Ref. [1], which is still very different. I believe this work can shed light on the training dynamics of GD in this setting.


References:

[1] - https://arxiv.org/pdf/2401.10791

**Weaknesses:**

The main weakness of this work is its **scope**. The work studies in great detail a very specific architecture, but it is unclear how interesting their conclusions are for either theorists or practitioners. Additionally, their numerical/toy example is extremely simple, and it is a bit hard to see how their results extend to more complicated settings, even though they discuss some of these aspects in their limitations section. It would be useful to include more complicated examples, even in the single hidden layer case, as well as contrast these results with cases in which the activations are not ReLU-like, i.e., commenting on how these results break down.

**Questions:**

Comments and questions below:

1) L52: “studied by under various…” incoherent sentence.

2) L51: “activation” should be “activations” I assume.

3) In Sec. 3.3, the authors consider an arbitrary dataset and a general empirical loss function. The analysis does not depend on these details and is therefore truly a property of the network and the initialization, and so these results should hold for any task. Is this correct?

4) The experimental setup in Sec. 6 is incomplete - the loss is never specified (MSE, CE?), it would be nice  to see (empirically) that the results do not depend on it.

5) Can the authors comment on the relation of their work and Refs. [1,2]?

6) What happens to the topological structure if we consider the discrete GD instead of GF? the results should hold for small learning rate (as shown numerically), but there is a possible catapult regime (Ref. [3]) at large learning rate, will the paths not be restricted to connected parts in this case? could the catapult effect cause a transition between disconnected effective manifolds?

[1] https://arxiv.org/pdf/2401.10791

[2] https://arxiv.org/pdf/2402.05626

[3] https://arxiv.org/abs/2003.02218

**Limitations:**

The authors discuss their limitations in a dedicated section.

---

> ### Author Rebuttal · Authors · 2024-08-06
>
> Dear Colleague, thank you very much for your useful and insightful review.
> ___
> **Questions.**
>
> **Q1,Q2.**
>
> Thank you for the corrections.
>
> **Q3.**
>
> That is correct. Particular activations or datasets may induce other symmetries but our results are independent of their choice.
>
> **Q4.**
>
> Thank you for pointing out the lack of specificity about the loss function, which we will fix in the final version of the paper. We indeed used the MSE loss. You can appreciate the generality w.r.t. the loss function in the experiment described in part 2 of the general rebuttal where the BCE loss is employed.
>
> **Q5.**
> The main relation of [1] with our work is in their Lemma 1 (balancing in GF) and the subsequent observation on the constant sign of the the weight on the outgoing edge, when $e=1$, which is the same observation proved in our Corollary 3. Nevertheless, while it is not difficult to prove corollary 3 as a consequence of the balance equations in the case e=1, our topological approach not only shows that the obstruction happens if and only if e=1, but also it explains in precise mathematical terms why the obstruction appears or not.
>
> In [2], the authors study one-layer networks with homogeneous activation functions and, in the case of MSE loss, they derive a condition for a stationary point to be a local minimum.
> This condition prescribes that there must be no *escape neurons* which, intuitively, provide escape directions from a saddle point.
> In the case of a scalar output ($e=1$), they prove that this condition is necessary and sufficient and that escape neurons are characterized by having $W^{(2)}_k = 0$.
> The general goals of the paper are different from ours, as we don't make any assumption on the loss function and, therefore, we don't aim at characterizing its local or global minima. However, it would be interesting to study the relations between the two works, in particular how our *pathological neurons* relate to the escape neurons.
>
> **Q6.**
>
> Please refer to part 3 of the general rebuttal, where we show one example of gradient descent circumventing the obstruction.
> At the moment, it is difficult to know if this is due to the catapult mechanism, and we think that understanding it is an interesting future research direction.
> ___
>
> **Weaknesses**
>
> **W1.** *The main weakness of this work is its scope. The work studies in great detail a very specific architecture, but it is unclear how interesting their conclusions are for either theorists or practitioners.*
>
> This shallow network as a model has limitations, but our work could become a theoretical starting point for studying other architectures in a similar way. Indeed, single-layer ReLU neural networks are commonly embedded as sub-networks of architectures like CNNs, which could show obstructions of the same kind studied here.
> The implications are not direct on the practical side, but initialization schemes could benefit from circumventing the possibility of obstruction by having $c_k>0$ for every hidden neuron.
>
> **W2.** *Additionally, their numerical/toy example is extremely simple, and it is a bit hard to see how their results extend to more complicated settings, even though they discuss some of these aspects in their limitations section. It would be useful to include more complicated examples, even in the single hidden layer case, as well as contrast these results with cases in which the activations are not ReLU-like, i.e., commenting on how these results break down.*
>
> Please refer to part 2 of the general rebuttal, where we describe a further, more realistic experiment.

---

> > ### Comment · Reviewer_9QVd · 2024-08-12
> > **Reply to the Authors**
> >
> > I thank the authors for their detailed reply, as well as their global reply and further experiments.
> >
> > I believe the paper should be accepted, but due to its limited scope, and after reading other reviews, including that of Reviewer JxYP, which addressed regularization and the "building block" argument etc., I believe that a higher score is not warranted.
> >
> > I do recommend that the authors delve deeper into the finite learning rate in future works, and especially the lr=0.533 example that was found, to understand how the obstruction is circumvented as a theoretically motivated reason for using large learning rates.

---

### Official Review · Reviewer_hKtn · 2024-07-13

**Soundness:** 3
**Presentation:** 3
**Contribution:** 2
**Rating:** 6
**Confidence:** 1

**Summary:**

In this paper, authors analyze the performance of gradient-descent optimization over a two-layer neural network. Authors reveal the presence of obstructions in the loss landscape and explore their topology. Finally, they identify the cause of those optimization obstructions in the so-called `pathological’ neurons—neurons that can’t change the sign of their output weight.

**Strengths:**

1) An interesting and novel approach.

2) A step up from well studied gradient descent optimization of one hidden-layer networks.

3) Solid theoretical background and strict mathematical proofs of main results.

**Weaknesses:**

1) This paper identifies pathological neurons that disrupt the performance of gradient descent over ReLU activations; however, nothing is said about how such obstructions can be avoided or how their effect can be mitigated.

2) The scope of the research is limited to neural networks with two hidden layers.

3) The possible benefits for further research and practical applications of the achieved results are somewhat vague.

**Questions:**

See Weaknesses section.

**Limitations:**

Yes, the limitations are addressed. This is a purely theoretic work, dealing with a somewhat simplified example of a two layer fully-connected neural network.

---

> ### Author Rebuttal · Authors · 2024-08-06
>
> Dear Colleague, thank you very much for your useful and insightful review.
> ___
> **Weaknesses.**
>
> **W1.**
>
> As we mention in lines  245 and 246, one can easily control the initialization to have $c_k\geq 0$ for every hidden neuron by employing, for example, Proposition 4. Once that holds we have that the invariant set is connected and the topological obstruction is avoided. Besides, please take a look at part 1 of the general rebuttal, where we address the probability of having obstructions under some of the more common initialization schemes.
>
> **W2.**
>
> We agree that being restricted to two-layer neural networks is the main limitation of our work. We are currently working on extending the results to multiple layers, and we are obtaining encouraging results. However, we believe that the two-layer case provides an elegant and important proof of concept on the possibility of studying topological obstructions to learning. Moreover, while most architectures are not two-layer neural networks, most of them contain two-layer networks as building blocks, and therefore, we can conjecture that obstructions like the ones studied in our work can occur even in that case.
>
> **W3.**
>
> While ours is a theoretical work, we believe that the main practical implication is that, when training neural networks with scalar output, one should check the values of $c_k$ for each hidden neuron and, if possible, choosing an initialization that ensures $c_k>0$. In that way, in fact, we can be sure that no topological obstruction to learning of this kind can occur.
>
> The future research directions, in our opinion, are many and interesting. First, the extension to the multilayer case is the most natural one and can also provide insights into other architectures, like CNNs, which can be seen as subnetworks of MLPs. Second, the literature provides us with other symmetries, like the scaling symmetry given by batch-norm and the translation symmetry given by adding a softmax before the loss. These symmetries induce constraints which can be studied with our framework.

---

> > ### Comment · Reviewer_hKtn · 2024-08-11
> > **Thank you for the clarifications**
> >
> > Thank you for answering my questions and providing clarifications; I have also carefully read the reviews from other reviewers (and rebuttals to them). I found this paper quite interesting and I lean towards thinking it worthy of acceptance. Unfortunately, my prior experience in the optimization theory is rather limited, and I will hold on my current score for this paper (6).

---

### Official Review · Reviewer_n2cB · 2024-07-13

**Soundness:** 3
**Presentation:** 3
**Contribution:** 2
**Rating:** 6
**Confidence:** 4

**Summary:**

This paper studies the topological obstruction in the loss landscape of two-layer ReLU neural networks under gradient flow. Using conserved quantities in gradient flow, the authors define invariant sets, to which gradient flow trajectories are constrained. They then show that when the input or output of the network is a scalar, the invariant set may have more than one connected components depending on initialization. This leads to a topological obstruction, as gradient flow cannot reach a global minimum if parameters are initialized in the component that does not contain one. After taking scaling and permutation symmetry into account, the number of effective components scales linearly with the number of pathological neurons, which are neurons that cannot change the sign of their output weight. On a toy example with two hidden neurons, the authors fully characterize the invariant sets and show theoretically and empirically that gradient flows that do not start in the same effective component as the global minimum cannot reach the global minimum.

**Strengths:**

- This paper takes a novel approach of studying the topology of loss landscapes by examining invariant sets. While many work studies the loss landscape by proving topological properties of sub-level sets, this paper pursues an orthogonal direction by analyzing the topological properties of the set of reachable parameters by training trajectories. This approach combines the study of loss landscape and optimization dynamics, which leads to the original discovery of topological obstructions to the training of ReLU networks.
- The paper provides concrete examples where gradient flows from certain initializations cannot converge to a global minimum and gives practical suggestions on how to choose initializations to avoid this issue, at least for two-layer ReLU networks.
- The results on when invariant sets can be disconnected and the number of effective components provide valuable information on the loss landscape. These results could potentially lead to a new line of work on the trainability of various architectures under common optimization algorithms.
- The paper is very well written. Setups and mathematical concepts are explained in an accessible way without losing rigor. The theorems and their implications are explained clearly. The visualizations (Figure 1-3) are informative and provide clear intuitions. The toy example in Section 6 is simple and effectively demonstrates a case where topological obstruction prevents gradient flow from reaching the global minimum.

**Weaknesses:**

The scope of the analysis, which is restricted to two layer ReLU neural network and gradient flow, is rather limited.
- While ReLU is a popular activation function, two-layer networks are rare in today’s machine learning models. The results in the paper do not readily extend to deeper networks, where invariant sets no longer factorize easily into product spaces.
- Since SGD is known to have an implicit bias characterized by a drift of $c$, training trajectories in practical settings (with non-infinitesimal learning rate) will not stay in the same invariant set. Even if they stay close to an invariant set throughout training, the noise in each step may take the parameters to a different component.

If extending the analysis beyond this setup is challenging, additional experiments on different settings could also help illustrate that the topological obstruction is a realistic hindrance in practice. For example, for deeper ReLU networks, it might be possible to investigate empirically whether there exist initializations that lead to similar obstructions as observed in Figure 3c.

**Questions:**

- For ReLU networks with one hidden layer, does every invariant set contain a global minimum? If not, then even if an invariant set is connected, it is not guaranteed that all gradient flows starting from points on this set can reach a global minimum. This is not the type of obstruction considered in the paper, but I am curious about whether there are other considerations when choosing the initialization, besides ensuring the connectedness of the invariant set.
- Assume that an invariant set contains a global minimum. Do all gradient flows in this invariant set converge to a global minimum?
- For a wider two-layer ReLU network, using common initialization methods (such as Xavier initialization), what is the probability that the invariant set has more than one effective components? What is the probability that the initialization falls in the component that does not contain a global minimum? Is it nonzero?
- Do the rest of Betti numbers, other than $\beta_0$, provide useful information about the loss landscape and the training process?

**Minor issues / suggestions**
- Line 135-136: While allowing a finite sequence of actions is arguably more intuitive, it might suffice to just define observationally equivalence as being related by a composition of one $T$ and one $P$. As a consequence of Lemma 3 in the appendix, for any finite sequence of T and P, there exists a sequence of T and P with length at most 2 that produces the equivalent transformation.
- In Equation 11, is the second expression missing a factor of 2?
- The sentence in line 483 is not grammatically correct.

**Limitations:**

The authors are upfront about the limitations and include a limitation section in the appendix.

---

> ### Author Rebuttal · Authors · 2024-08-06
>
> Dear Colleague, thank you very much for your useful and insightful review.
> ___
> **Weaknesses.**
>
> **W1.**
>
> It is true that our results are limited to one-layer neural networks but we believe that, even if they cannot be generalized to the multi-layer case, they could be interesting when studying architectures which employ one-layer NNs as building blocks (like CNNs).
>
> **W2.**
>
> Indeed, both SGD and GD with finite learning rate will drift away from the invariant set and, if $c_k$ is small enough, we find that this drift can push the trajectory to circumvent the obstruction (see part 3 of the general rebuttal). We stress, however, that from the point of view of our analysis, there are no differences between continuous time GD and continuous time SGD. This comes from the fact that the latter can be seen as a GF with respect to a different loss which depends only on a subset of the dataset (a mini batch) and thus is also invariant to the same rescaling action.
>
> ___
> **Questions.**
>
> **Q1.**
>
> Thank you for the interesting question. The answer is to be found in Proposition 4. What it tells us is that, for every non-degenerate parameter $\theta$, there is a rescaling that takes $\theta$ to an observationally equivalent one. This means that every invariant set contains copies of all (non-degenerate) parameters and, therefore, the global minimum too.
>
> **Q2.**
>
> Unfortunately, we cannot answer this question a priori as the actual GF trajectory will depend on the specifics of the loss function, i.e., its functional form and the dataset used. By staying completely general, we cannot exclude that the training will get stuck in a local minimum, even if the initialization is in the right component. What we can say is that if $\mathcal{H}(c)$ is not connected, then, surely, a global optimum in another component cannot be reached.
>
> **Q3.**
>
> For the same reasons described in the reply above, we cannot, staying completely general, quantify whether a global minimum will be reached. We can, however, compute some statistics about the topological properties of the invariant set under common initialization schemes. Please see part 1 of the general rebuttal for a thorough answer about the probability of connectedness.
>
> **Q4.**
>
> This is a very interesting question for which we do not have a clear answer yet. While it is clear how connectedness can affect the learning trajectory, it is not clear if and how higher-order holes can affect it.
>
> **Minor issues/suggestions.**
>
> Thank you for pointing out these mistakes, which we will amend in the camera-ready proof, hopefully.

---

> > ### Comment · Reviewer_n2cB · 2024-08-11
> > **Thanks for response**
> >
> > Thank you for addressing my comments and questions. I do not have further questions and will maintain my positive rating.

---

### Official Review · Reviewer_JxYP · 2024-07-14

**Soundness:** 4
**Presentation:** 4
**Contribution:** 3
**Rating:** 7
**Confidence:** 4

**Summary:**

Summary of the paper:

* The paper undertakes a theoretical 'reachability' analysis for neural
  networks trained with gradient flow, identifying 'obstructions' separating
  certain regions of parameter space from each other.
* The setting is one-hidden-layer networks with homogeneous activation
  functions, including ReLUs. The main paper considers the biasless case but
  the authors extend their results to the case with biases in the appendices,
  with only minor modifications to the results.
* The authors consider training such networks with gradient flow and identify
  that regardless of the (output-dependent) loss function that gradient flow
  conserves a certain norm of the weights confining the training trajectory
  to a subset of the parameter space they call the 'invariant set' which is a
  product of hyperquadrics associated with each hidden unit in the network.
* Through topological analysis the authors derive the Poincaré polynomial of
  such invariant sets and in turn their zeroth Betti number, in effect
  counting the number of connected components in the invariant set.
  * These components are of interest because if gradient flow is initialised
    in one component it is impossible for the training trajectory to find its
    way to another, and training is limited to the best solution available in
    the initial component.
  * In this setting the zeroth Betti number (the number of connected
    components) is only one for networks with multidimensional inputs and
    outputs, indicating no topological obstructions.
  * In contrast, for networks with scalar inputs or outputs, it is
    exponential in the number of hidden units (with a base depending on the
    network initialisation), indicating many topological obstructions.
  * (For networks with biases, the space is connected if and only if the
    network has a multidimensional output).
* The authors then revise their count to take rescaling and permutation
  symmetries into consideration.
  * The reasoning is that if gradient flow can only explore its initial
    connected component, this is only a problem when the component is
    actually missing network functions rather than when it is merely missing
    some (but not all) implementations of those functions. Perhaps only one
    reachable implementation of the target function is needed.
  * The result is that there are fewer 'effectively' connected components,
    linear rather than exponential in the number of hidden units (with
    constants again depending on the initialisation).
  * However, the point is that there are still some topological obstructions
    to worry about.
* The authors provide a clear 'proof of concept' of their work in the form of
  a numerical experiment simulating gradient flow by learning a small neural
  network with gradient descent using a small learning rate. The results
  clearly show that the learning trajectory is confined to the invariant set
  and obstructed from reaching the global minimum as predicted.

I wrote a bit of a long review, so here is also a summary of my review:

* The main contribution, demonstrating the existence of topological
  obstructions to gradient flow, appears novel, interesting, and an important
  contribution to our understanding of the structure of parameter space.
* I also think the paper is exceptionally well written and clear and the
  analysis is elegant. I think the authors have discussed the limitations of
  the work well. I am left with only a small number of questions.
* My main concern is that the results show topological obstructions only
  occur under some strong assumptions, and in fact they do not arise in
  more situations. I find this somewhat at odds with the framing of the
  contribution.

Overall, I think the work is worthy of acceptance because it is elegant,
interesting, informative, and thought provoking, but I maintain concerns
about whether the framing is accurate.

**Strengths:**

I thank the authors for submitting their elegant and well-presented theoretical
analysis exploring an important topic.

* Topological analysis of gradient flow is an interesting and apparently
  novel approach that stands to make a meaningful contribution to the field's
  theoretical understanding of the 'global' structure of parameter space.

* The work clearly exhibits 'topological obstructions' in a restricted
  setting, which is a thought-provoking phenomenon (though I have some
  reservations about its relevance in practical settings, see below).

* I appreciate the elegant and rigorous application of powerful, established
  tools from the field of topology. The theoretical results are also
  supported with a clear, small-scale proof-of-concept numerical experiment.

* The presentation of the theory and experiments is very clear and
  accessible. The figures are carefully designed and informative. The
  technical details of the framework and results are complete and made the
  paper quite accessible to me, even though I have limited knowledge of
  topology (the primer in the appendix was appreciated).

* Clear, thorough acknowledgement of related and prior work in the dedicated
  section and throughout the paper.

**Weaknesses:**

The authors have convincingly demonstrated the existence of topological
obstructions under the stated assumptions. My main concern about this work is
the relevance of topological obstructions to deep learning practice.

1. I am not so concerned about the restriction to single-hidden-layer
   networks. The results may not generalise to the multi-layer case, or
   perhaps they will, it seems worth looking into with future work. However,
   I am concerned that *even in the single-hidden-layer setting* the main
   disconnectedness result holds only for networks with scalar output. The
   authors clearly show that networks with multidimensional outputs do not
   show obstructions.

    With this in mind, do the authors believe that the topological
    obstructions they have identified are relevant to deep learning practice?
    If so, can they clarify this point?

2. Furthermore, I have concerns about the gap between gradient flow and
   discrete gradient descent training. The authors explain that discretised
   gradient methods are not confined to the same invariant sets as gradient
   flow. It seems possible that this means discrete gradient methods count
   circumvent topological obstructions, even if they exist in practical
   settings.

    1. Do the authors conjecture that the identified obstructions have
       implications for gradient descent training?

    2. If the authors would support such an 'obstruction hypothesis', I would
       invite them to consider explicitly stating the hypothesis in their
       introduction and/or conclusion.

    3. The authors could also consider extending their simple numerical
       experiments to explore whether and to what extent the demonstrated
       obstruction affects training with larger learning rates or
       momentum-based training methods.

Pending the authors' clarification of these points, the framing of the work
is somewhat confusing to me. It appears to me that the authors have shown
that topological obstructions to gradient flow are actually quite a
restricted phenomenon. An alternative framing for the paper's results would
be to say that they show that deep learning is often *not* held back by
topological obstructions. Could the authors comment on this alternative
framing?

I don't think this concern is fatal to the paper. The topological analysis
approach and most of the results are still informative and worth sharing even
if the topological obstructions themselves end up not arising in more
practical settings.

**Questions:**

1. The authors have presented a clear interpretation of the source of the
   topological obstruction in terms of the relationship between each
   unit's weights (lines 232 onwards). Do the authors have a similarly
   enlightening interpretation of the breakdown of the topological
   obstructions in the case with multidimensional outputs?

2. Another assumption of the work is that the loss function depends only on
   the network weights via the output of the network. Am I correct in
   understanding that if the loss function includes a regularisation term
   that depends on, for example, an $L_p$ norm of the weights, then this
   assumption would be violated and the analysis would not hold because
   gradient flow would no longer always be perpendicular to the invariant
   sets? This seems fine but might be worth acknowledging.

3. Appendix E extends the analysis to networks including biases, but stops
   after deriving the zeroth Betti number in this setting. Could you please
   clarify whether the analysis of section 5 is unchanged in this setting?

4. The contribution list does not mention the 'proof of concept' numerical
   experiments. It seems to me that these experiments are a reasonably
   important part of the paper's contribution and its evaluation to warrant
   mentioning in the introduction (as well as the abstract).

5. Line 118: The authors describe the transformations as sending parameters
   to different but observationally equivalent parameters. It is typically
   true that the parameter produced is different but for some parameters or
   for some transformations the transformations don't change the parameter.

**Limitations:**

The paper accurately describes the contribution including its limitations.
These limitations are adequately and carefully disclosed in multiple places
in the main text and discussed in detail in appendix F.

1. One main limitation is the simplicity of the architecture, with only one
   hidden layer. The authors discuss barriers to extending the work to the
   multi-layer case, namely the interaction between layers, which is left to
   future work.
2. A second main limitation is the fundamental assumption of gradient flow.
   The authors explain that the conservation laws that are crucial to their
   analysis do not hold for discretised gradient descent.
3. A third main limitation is that the topological obstructions only exist in
   the case of networks with a scalar output. This is clearly acknowledged in
   the abstract and throughout the paper, but I think the authors could have
   more thoroughly discussed the implications of this restriction, namely
   that in many cases we should not expect topological constructions at all.

If the work is to be accepted I would encourage the authors to include the
valuable discussion in appendix F in the main text, and to extend this
discussion along the lines of (3).

---

> ### Author Rebuttal · Authors · 2024-08-06
>
> Dear Colleague, thank you very much for your useful and insightful review.
>
> ---
>
> **Weaknesses**
>
> **W1.**
>
> At the moment it is hard to foresee if and how this can be relevant to deep learning practice as the results need to be extended to more general settings.
> Provided the same holds also for multilayer networks, then we could see this obstruction being relevant as MLPs are often employed as building blocks of commonly used architectures like CNNs.
>
> Focusing on the single-layer case, we think that the impact of obstruction is also related to its width and the particular task at hand: if the NN is large and the task is not hard, it will probably be able to solve it without employing its pathological neurons, provided they are not too many.
>
> Assuming, however, that the network is not too large, there are specific settings where obstructions could be a hindrance.
> As you highlight, the case of scalar outputs is a limited scenario that is nonetheless relevant in tasks like binary classifications.
> As you can see from part 2 of the general rebuttal (Fig. 2 in the extra pdf file), we observe that, by training with GD, increasing the number of pathological neurons at initialization reduces the average classification performance.
>
> **W2.1**
>
> The precise relation with finite-step size GD is an interesting direction that we will certainly explore in the future. From the simple preliminary experiments we performed, we tend to see that, even when the trajectories drift away from the invariant set, such drift is usually not able to push them over the obstruction. The experiment described in part 3 of the general rebuttal shows, however, that this behavior can occur for some values of the learning rate, which are not too low nor too high. Our conjecture is that, in general, this is more likely to happen when the initial value of $c_k$ is negative but small so that the connected components are close to one another.
>
> **W2.2**
>
> Given that all of our analysis is based on the loss functions' invariance w.r.t. to rescaling and the parameters evolving through GF, we think that the addition of momentum would break the setup and result in trajectories that are not constrained on $\mathcal{H}(c)$.
>
> Regarding the effects of higher learning rates, please refer to part 3 of the general rebuttal, where we show a simple experiment in which the trajectories circumvent the obstruction.
>
> **W2.3.**
>
> We agree that, from this first set of results, it seems that the topological obstruction induced by symmetries is a restricted phenomenon. However, the fact that there is the possibility of a fundamental obstruction to training is not something that is commonly known or expected by the community. Therefore, we believe that the framing we gave to the paper is justified, as it serves to highlight this possibility. Moreover, while our analysis clearly identifies obstructions in some cases, it is, in principle, hard to exclude that extending the results to more general settings will not uncover other obstructions of a similar kind. Stating that "deep learning is often not held back by topological obstructions" would, therefore, be a bit optimistic at this point, even if the probabilistic analysis of part 1 of the general rebuttal seems to corroborate this conjecture for two-layer networks.
>
> ---
>
> **Questions.**
>
> **Q1.**
>
> Yes, we can give some intuition on why there is no obstruction for multiple outputs. Let us first consider a single hidden neuron, with incoming weights $w^{(1)}\in\mathbb{R}^d$ and one single output $w^{(2)}\in\mathbb{R}^d$. If the neuron is pathological, we have that $||w^{(1)}||^2_2 - (w^{(2)})^2 = c$ with $c<0$. Since the weights $w(t)$ are continuous, for $w^{(2)}$ to change sign, its value needs to pass through 0 but, under the condition above, this cannot happen as $(w^{(2)})^2 = ||w^{(1)}||^2_2 - c >0 \implies w^{(2)}\neq 0$. Consider now multiple outputs $w^{(2)}\in\mathbb{R}^e$, resulting in the condition being  $||w^{(1)}||^2_2 - ||w^{(2)}||^2_2 = c$. Now, any component of $w^{(2)}$ can change sign by passing through 0 because the other components can compensate for it by increasing their magnitude to keep the condition satisfied.
>
> **Q2.**
>
> Yes, this is correct. If we add $L_p$ regularization, then the loss function is no longer invariant to the action of the rescaling transformation, and thus, the learning trajectory is, in general, not constrained anymore to lie on $\mathcal{H}(c)$.
>
> **Q3.**
>
> Yes, all the results shown in Section 5 also hold when biases are taken into account. This comes from the fact that, from the formal point of view, biases are the same as the weights of $W^{(1)}$. We can thus treat a two-layer NN with biases as a biasless two-layer NN with $d+1$ instead of $d$ inputs. If we extend the permutation action of Eq. (4) to also permute biases, the results will be left practically unchanged except for the first hypothesis of Theorem 2, which will now become $d\geq 1$ instead of $d>1$.
>
> **Q4.**
>
> Thank you for the comment. If accepted, we will add it to the final version of the paper.
>
> **Q5.**
>
> Thank you for the observation. We consider rephrasing that sentence to make it more precise, e.g., "We describe two kinds of transformations whose orbits are made of observationally equivalent parameters only.

---

> > ### Comment · Reviewer_JxYP · 2024-08-11
> > **Thanks for the clarification**
> >
> > I thank the authors for their clarifying rebuttals. I especially thank the authors for their intuitive explanation of the resolution of the obstruction in the multi-output case. I invite them to include this and their other clarifications in future revisions of the paper. The three new experiments are also a very welcome addition to the submission.
> >
> > At the moment I am maintaining my recommendation. I retain some of my concerns about the framing and relevance of the results. I am about to post a top-level comment on this topic since concerns about relevance were also raised by a few other reviewers.

---

### Author Rebuttal · Authors · 2024-08-06

We thank the reviewers for their insightful and thorough assessments of our work.

Prompted by some of the reviewers' questions, we performed three additional experiments to clarify some of the points raised.

### **1. Probability of obstruction**
Let us consider the following question: *what is the probability of having a disconnected invariant set given a realistic initialization?*

Consider a one-layer neural network with $e=1$ and assume that the weights are sampled independently of one another from a normal distribution
$W^{(1)}\_{ki}\sim\mathcal{N}(0,\sigma_1^2)\ \forall k,i$ and $W^{(2)}\_{k}\sim\mathcal{N}(0,\sigma_2^2)\ \forall k,j$.

From Corollary 2, we know that the invariant set $\mathcal{H}(c)$ will be disconnected if and only if there exists a hidden neuron satisfying the following $\sum_{i=1}^d (W^{(1)}_{ki})^2 < (W^{(2)}_k)^2$.

Given the independence of weight sampling, this probability can be computed as
$\mathbb{P}[\text{obstruction}] = 1-\mathbb{P}[\sum_{i=1}^d (W^{(1)}_{ki})^2 > (W^{(2)}_k)^2]^l$,

 which it is not hard to show to be equal to $1-(F_{1,d}(d\sigma_1^2/\sigma_2^2))^l$, where $F$ is the cdf of the Fisher-Snedecor distribution.

Having obtained this general expression, we can specify it to two common initialization schemes.

- We obtain *Kaiming initialization* [1] with $\sigma_1^2 = 2/d, \sigma_2^2 = 2/l$ resulting in $\mathbb{P}[\text{obstruction}] = 1 - F_{1,d}(l)^l$;
- We obtain *Xavier normal initialization* [2] with $\sigma_1^2 = 2/(d+l), \sigma_2^2 = 2/(1+l)$ resulting in $\mathbb{P}[\text{obstruction}] = 1 - F_{1,d}(\frac{d+ld}{d+l})^l$.


We plot these two expressions in Figure 1 of the extra file we uploaded.
We can clearly see how, for large values of $d$, the probability of obstruction quickly falls to 0 for any number of hidden neurons.
Instead, for small values of $d$, we see an opposite trend: the probability of disconnectedness grows with $l$.
Moreover, it is interesting to notice that the region of high obstruction probability is much larger for Xavier initialization than for Kaiming initialization, further showing that the latter is preferred when working with ReLU networks.

As a complement, we also observe that, by using the binomial distribution, it is possible to show that the probability of having $2^B$ disconnected regions, assuming the same sampling scheme described above, is  ${ l \choose B}p^B p^{l-B}$, where $p=1-F_{1,d}(z)$, with $z$ the opportune argument depending on the variances or the layers' sizes.



### **2. A more realistic experiment**
We present here a further experiment to show how the topological obstruction presented in the paper can be a hindrance in a more realistic setting.
We consider a simple binary classification task on the well-known  *breast cancer* dataset [3] which we try to solve by fitting one-layer ReLU neural networks trained to minimize BCE loss.
We vary the number of hidden neurons $l$ and, for each $l$, we change the number of neurons $k$ such that $c_k>0$ from 1 to $l$.
By repeating the experiment 100 times, we can show how the model's average performance changes when the degree of disconnectedness of its invariant set is varied.
The result, on the left panel of Fig. 2 clearly shows the presence of a "gradient" in performance, where increasing the number of positive $c_k$ (non-pathological neurons) tends to decrease the average value of the test loss after training.

### **3. Finite step size**
The results presented in our work hold when the network is trained with continuous-time GF so it is natural to wonder what changes when we consider finite-step size gradient descent.
While a formal or thorough numerical analysis is outside the scope of this work, we show in a simple setting how *it is possible* for a large enough step size to make a trajectory abandon the initial $\mathcal{H}(c)$ and circumvent the obstruction.
We consider the same numerical setup of Section 6 and initialize the parameters on the wrong component of a $\mathcal{H}(c)$ with $c_1=c_2=-0.03$ so that it results in disconnection while also being small enough to make the component close and facilitate the trajectory's escape.
In Fig. 3 of the supplementary pdf, we see how raising the learning rate high enough can change the $c_k$ of the first neuron, pushing it over the obstruction and allowing it to reach the global minimum close to the other component.

- [1] *Delving deep into rectifiers: Surpassing human-level performance on imagenet classification -  He, K., Zhang, X., Ren, S. \& Sun, J. (2015).*
- [2] *Understanding the difficulty of training deep feedforward neural networks - Glorot, X. \& Bengio, Y. (2010).*
- [3] *Breast Cancer Wisconsin (Diagnostic). UCI Machine Learning Repository - Wolberg, W., Mangasarian, O., Street, N. \&  Street, W. (1995).*

---

### Comment · Reviewer_JxYP · 2024-08-11
**Relevance of topological obstructions to deep learning practice [1/2]**

It appears that the current consensus among reviewers is that this paper should be (weakly) accepted, with the primary remaining concern being about the relevance of the phenomenon of topological obstructions in practice (noting this is my impression and I don't speak for the other reviewers).

On the question of relevance I have a couple of follow-up questions after reading the reviews and rebuttals and studying the new gradient descent experiments. I organise my questions by two key points of departure between the theoretical setting where topological obstructions found by the authors and those settings common in deep learning practice: on the learning algorithm (gradient flow) and on the architecture (scalar-output single-hidden-layer ReLU networks). I conclude with some suggestions on the paper's framing.

### Learning algorithms

**1. Characterising when discrete GD bridges obstructions:** The additional experiments show that the phenomenon of discrete gradient descent training bridging the obstruction "can occur for some values of the learning rate, which are not too low nor too high" [from a rebuttal].
I would like to question the basis of this "not to low nor too high" characterisation.

In the attachment I see that the `lr=0.533` run clearly bridges the obstruction in terms of loss and parameter-space trajectory. The `lr=0.6` run appears not to have bridged the obstruction (though I can't actually see where it went in the parameter space?), and this must be the basis for "not too low nor too high", but this is the only higher learning rate tested. I don't know that this is sufficient basis for characterising the bridging phenomena in this way.

**2. On regularisation:** In my review I noted that learning with weight-based regularisation would violate the assumptions of the analysis, and the authors agreed with this observation.

This difference would appear to me to be more severe than the difference between gradient flow and (stochastic) gradient descent, since it is not just about discretisation---regularised parameter updates may not even be *locally* orthogonal to the invariant set.

This limitations are implicit in the assumptions listed throughout the analysis, but I would invite the authors to explicitly acknowledge this limitation at the level of the discussion of appendix F (alongside their existing acknowledgement of the difference with momentum-based methods).

### Architectures

**3. On biases:** I was not completely convinced by the authors' reply to my question about the case with biases. It's true that networks with biases are the same as networks with additional input units, but those additional input units have to have their inputs restricted to a constant function. Thus analysis of network functions including, e.g., invariance, has to proceed in a subspace of the expanded architecture's function space. In general such a restriction can create new symmetries.

I think this probably doesn't undermine the analysis, since the analysis is mainly based on the existence of certain symmetries which will not be removed by adding additional restricted dimensions. I would need to think about it more carefully to be sure.

I would like to ask, have the authors taken this complexity into account in the claims in the paper that the results generalise to networks with biases?

**4. On single-hidden-layer networks as building blocks:** A couple of times in the rebuttals the authors said that while single-hidden-layer networks are not commonly used in practice as a stand-alone architecture, they are commonly used as building blocks of other architectures. The authors cite CNNs as examples and I might add their use in transformer blocks and that an MLP can be viewed as a composition of
single-hidden-layer networks in some sense.

However, I wanted to point out that in such cases typically the building block layers have multiple outputs (with the exception of a final layer in an MLP or CNN used for binary classification). The authors have shown that such layers *would not* unilaterally suffer topological obstructions. Therefore, while the authors' analysis could be informative in studying architectures composed of hidden layers, their key finding won't necessarily generalise (there would have to be some other component of the architecture that restores the topological obstruction to this layer or introduces its own obstruction). In the case of networks with a final layer with a scalar output, the final layer may have a topological obstruction, as long as the interaction between this and prior layers does not allow it to be circumvented.

I understand that the definitive answer would have to be left to future work, but I wonder, would the authors agree with this preliminary assessment of the prospects of that future work?

[Part 1 of 2]

---

> ### Comment · Reviewer_JxYP · 2024-08-11
> **Relevance of topological obstructions to deep learning practice [2/2]**
>
> ### Framing
>
> **5. On the position of the limitations section:** I think the differences of architecture and learning algorithm between the setting studied in this paper and settings of relevance in deep learning practice are substantial and deserve to be explicitly addressed within the main text of this paper if it is to be accepted.
>
> I think the authors have acknowledged these differences and their appendix F does a good job of summarising them. I would repeat the suggestion from my review, that the authors should consider promoting this material from appendix F into the main text.
>
> Since my review the authors have also presented a page worth of additional figures, and, also accounting for the space required to describe these experiments, they may face a shortage of space if they were to attempt to put all of this material into a 10-page version of the submission along with the discussion of limitations. However, I still think it is worth considering making a prominent place for a brief explicit acknowledgement of these limitations (e.g. in a named section or subsection) because this is one of the most important considerations for readers of the submitted work to be made aware of. (Of course, the authors are welcome to use their judgement to decide how to organise the final version of the paper, if accepted.)
>
> **6. Making explicit the open question of topological obstructions:** I questioned the authors' framing around topological obstructions, given that their results imply that these appear in quite restricted circumstances. I appreciate the authors' response which was as follows:
>
> > We agree that, from this first set of results, it seems that the topological obstruction induced by symmetries is a restricted phenomenon. However, the fact that there is the possibility of a fundamental obstruction to training is not something that is commonly known or expected by the community. Therefore, we believe that the framing we gave to the paper is justified, as it serves to highlight this possibility. Moreover, while our analysis clearly identifies obstructions in some cases, it is, in principle, hard to exclude that extending the results to more general settings will not uncover other obstructions of a similar kind.
>
> I agree that the finding of topological obstructions in a restricted setting is worth presenting to the community and worth further investigation.
>
> Upon taking another glance at the paper I can't fault the authors for anything they have said. They consistently carefully qualify every claim so as to avoid overstating their results. They have been upfront about the limitations of the direct implications of their results in their paper and in their rebuttals. In my opinion, the paper is a masterclass in precise communication of theoretical results.
>
> However I am left with a distinct impression that while the authors have not said anything wrong, something important remains *unsaid*---namely that the central motivation of their work is the hypothesis that topological obstructions may hinder training in practical deep learning. If I am not mistaken, this is never stated in the paper, even though it has got to be the strongest reason to share this work with the NeurIPS community (right?).
>
> I think the authors don't need to be afraid to make this claim, even though the answer lies in future work. Their results and experiments have comprehensively and elegantly answered this question in the limited setting of shallow ReLU networks. More than a toy model, their analysis made to look effortless with the powerful tools of topology has cut out an example for future work on this topic to follow.
>
> I therefore ask:
>
> 1. Do the authors agree that the reason people should care about their work is that topological obstructions may be relevant to deep learning with practical algorithms and architectures (or not, whatever the case may be, but it would be nice to know either way)?
> 2. Are the authors willing to state their answer to question (1) prominently in their paper (for example in the introduction or conclusion, for example alongside the discussion of limitations as per my previous point)?
>
> [Part 2 of 2]

---

> > ### Author Response · Authors · 2024-08-13
> > **Thank you for the comments**
> >
> > Dear Colleague, thank you for the positive assessment of our work and for the insightful comments.
> >
> > **1. Characterising when discrete GD bridges obstructions.**
> >
> > The preliminary experiment we performed, prompted by the reviewers' questions, was just meant to show that there are circumstances in which it is possible for gradient descent trajectories to circumvent the obstruction, and should not be taken as a complete characterization of the phenomenon. The argument "not too low nor too high" roughly meant that if the learning rate is too low, the trajectory closely approximates GF dynamics, which prevents it from escaping the invariant set. Conversely, if the learning rate is set too high, GD may exhibit erratic behavior.
> >
> > We think that analyzing the effect of discretization is a very interesting research direction which deserves to be treated in detail elsewhere. As for when the bridge phenomenon happens, an educated guess would be that the value of the learning rate should be compared with the minimal distance between the components of the invariant hyperquadric i.e. $2|c|$.
> >
> > **2. On regularisation.**
> >
> > Thank you for the comment. We will add this observation in the limitations section.
> >
> > **3. On biases.**
> >
> > The key point is that the work presented in this paper does not make any assumptions on the inputs, whether they can vary or they are constant (like for biases). As you can see in Eq. (26), the rescaling transformation (and also permutation) is not concerned with the inputs and acts on biases just like it acts on the weights of the first layer $W^{(1)}$. Therefore, weights and biases will be treated in exactly the same way and the results obtained still hold.
> >
> > **4. On single-hidden-layer networks as building blocks.**
> >
> > Your assessment is indeed correct. In the case of a network with a final two-layer ReLU module, the interaction with the prior layers would need to be taken into consideration for our topological analysis.
> >
> > **5. On the position of the limitations section.**
> >
> > We thank you for the suggestion. We agree that the limitation section, expanded with the observations prompted by the reviews, needs to be included in the main paper and, if accepted, we will do so in the final version.
> >
> > **6. Making explicit the open question of topological obstructions.**
> >
> > The driving interest behind our work is undoubtedly related to this "obstruction hypothesis", but we feel it is too early to pinpoint the potential practical applications.
> > We hope the community will find our paper intriguing because it studies a fundamental structural phenomenon of neural network architectures and, secondarily, because of the possibility of it leading to practical improvements to deep learning practices (if these obstructions are found to have an impact).
> >
> > On this second possibility, an insight we gather from the classification task presented in part 2 of the general rebuttal, is that the presence of only pathological neurons seems to hinder learning, even when the network's width is scaled. However, our probabilistic analysis shows that with common initialization schemes, the probability of creating a pathological neuron decreases rapidly with increased inner layer width. Therefore, the combination of specific initialization schemes and an abundance of hidden neurons (beyond the minimum required to solve a task) appears to make this obstruction unlikely in practice, which might explain why this phenomenon has not been observed.
> > This work describes a simple safeguard to avoid obstructions, which could, for instance, discourage looking for new initialization schemes that result in a proliferation of pathological neurons.
> >
> > Should the paper be accepted, we will include these observations and further clarify our motivations in the introduction and conclusions sections.

---

### Decision · Program_Chairs · 2024-09-25

**Decision:**

Accept (poster)

**Comment:**

The work studies the properties of the loss surfaces of two-layer neural networks with homogenous activations (e.g., ReLU). The authors prove that in such networks, the gradient flow trajectories lie on an invariant set, which can be factored as the product of quadric hypersurfaces. The authors derive the properties of such sets and, importantly, show that for neural networks with a single output neuron, such sets may be disconnected, which creates obstacles for gradient flow and makes it (sometimes) impossible to reach the global optimum.

The reviewers found the results of the paper to be interesting and inspiring, and the overall exposition to be very clear and accessible. The key concepts are well illustrated with the figures and experimental illustrations. The area chair after reading the paper is also on the positive side. The key weakness pointed out by the reviewers is the limited scope, as the paper (primarily) studies a particular setting (continuous gradient flow, two-layer MLP only, homogenous activations, primary focus on the scalar output, etc.). However, it is likely that some of the theoretical advancements developed in the current paper can be extended to more general cases of neural networks. As a result, this could help to broaden the theoretical understanding of neural networks and help to advance the field of deep learning from the practical side as well (e.g., NN initialization, optimization).

Overall, there has been a large amount of fruitful discussions of authors with the reviewers, with clear suggestions on how and what to further improve in the paper, in particular, to highlight the open problems and potential future research in this direction. Upon acceptance, the area chair requests the authors to spend sufficient time incorporating these discussions into the paper accordingly.